# Hbo1 and Msl complexes preserve differential compaction and H3K27me3 marking of active and inactive X chromosomes during mitosis

Dounia Djeghloul [1,2,9] ✉, Sherry Cheriyamkunnel[2,9], Bhavik Patel[3], Holger Kramer [4], Alex Montoya [5], Karen E. Brown[2], Chad Whilding[6], Tatyana B. Nesterova [1], Guifeng Wei [1], Neil Brockdorff [1], Iga Grządzielewska[7], Remzi Karayol[7], Asifa Akhtar [7], Matthias Merkenschlager [8] & Amanda G. Fisher [1,2] ✉

In mammals, chromosome-wide regulatory mechanisms ensure a balance of X-linked gene dosage between males (XY) and females (XX). In female cells, expression of genes from one of the two X chromosomes is curtailed, with selective accumulation of Xist-RNA, Xist-associated proteins, specific histone modifications (for example, H3K27me3) and Barr body formation observed throughout interphase. Here we show, using chromosome flow-sorting, that during mitosis, Xist-associated proteins dissociate from inactive X (Xi) chromosomes, while high levels of H3K27me3 and increased compaction of the Xi relative to active X (Xa), are retained. Proteomic comparison of mitotic Xi and Xa revealed that components of Hbo1 and Msl/Mof histone acetyltransferase complexes are significantly enriched on Xa as compared to Xi and autosomes. By contrast, inhibitors of histone acetylation co-enrich with Xi. Furthermore, inhibition of Hbo1 or deletion of Msl/Mof components functionally abolishes mitotic differences in H3K27me3 marking and chromosome compaction. These data uncover critical roles for acetylation pathways in preserving X chromosome properties during mitosis.

Chromosome-based sex determination, used by many different species, often necessitates co-evolution of dosage compensation mechanisms that fine-tune sex chromosome gene expression. In mammals, progressive erosion of genes on the male Y chromosome is thought to be compensated by adaptations leading to enhanced expression of X-linked genes, thereby correcting imbalances relative to other chromosome pairs[1–5]. These adaptations have driven the evolution of X chromosome inactivation (XCI), to correct for overexpression of X-linked genes in females (XX)[2,6–9]. Different species have evolved

distinct mechanisms, for example, in *Drosophila melanogaster*, genes on the X chromosome are upregulated specifically in males[7,8,10], and in *C. elegans*, partial repression of genes on both X chromosomes in hermaphrodites (XX) equalises expression levels relative to males (XY)[2]. In mammals, XCI occurs early in development in cells of female embryos[9] (reviewed in refs. 11–14). Studies in mouse have shown that this is achieved by a developmentally co-ordinated process of inactivation and reactivation of paternal X chromosomes early in zygotic development, followed by a second wave of random XCI (rXCI) where

one of the two X chromosomes in females is epigenetically silenced[15–22]. rXCI occurs as the embryo proper is generated creating mosaicism in female somatic tissue[23–25].

At the level of individual cells, distinctive transcriptional and epigenetic properties that are associated with individual X chromosomes (active or inactive) are maintained throughout rounds of cell division (reviewed in refs. 26,27). Microscopically, mammalian inactive (Xi) and active (Xa) X chromosomes are distinct. The Xi is heterochromatic, showing depletion of histone modifications associated with gene activity (for example, histone acetylation) and enrichment of histone modifications linked to gene repression (for example, histone H3 lysine 27 trimethylation, H3K27me3)[28]. The Xi also forms a distinctive Barr body structure in interphase, located close to the nuclear lamina[29,30]. By contrast, the Xa appears more similar to autosomes and is decondensed relative to the Xi[31–33] (reviewed in ref. 34). Orthogonal studies of mammalian chromosome morphology also revealed that Xi and Xa have distinctive structures and occupy different territories within interphase cells. Early investigations using confocal laser scanning and X-DNA probes to delineate specific domains, showed that Xi and Xa chromosomes had distinctive shapes and volumes[33,35]. Specifically, the territory of the Xa resembled autosomes of a similar size, while the Xi was more condensed. Subsequent chromosome conformation capture analyses have confirmed that in interphase the Xa and autosomes comprise several small topologically associating domains defined by boundary elements enriched for cohesin and CTCF, whereas, the Xi is bipartite with two large domains lacking cohesin or CTCF[36–38]. Although differences in X chromosome states are conserved between mouse and human[39], there is a paucity of structure comparisons between Xa and Xi in mitosis. This is important since cohesin and CTCF are routinely displaced from chromosome arms in mitosis and topologically associating domains are lost or substantially reduced[40–45].

At the molecular level, XCI is initiated by Xist, a 17 kb non-coding RNA derived from the Xi that coats and represses genes *in cis*[46–49]. Recent studies have established that Xist-mediated chromosome silencing is largely attributable to interactions between a critical element in Xist-RNA, the A-repeat, and the RNA binding protein (RBP) Spen, which brings about chromosome wide histone deacetylation by activating chromosome-bound Hdac3 (refs. 50–54). In parallel, B- and C-repeat elements in Xist-RNA are bound by the RBP HnRNPK, which in turn recruits Polycomb Repressive Complex 1 (PRC1)[55–58]. Recruitment of non-canonical PCGF5/3-PRC1 mono-ubiquitinylates histone H2A lysine 119, and this then recruits PRC2 that catalyses local H3K27me3 (ref. 59). In addition, other epigenetic features are linked to XCI maintenance including incorporation of macroH2A histone variant, increased DNA methylation, replication late in S-phase and propagation of H3K9 methylation[60–65].

We recently developed technologies to directly purify native metaphase chromosomes from dividing cells using flow cytometry[40,66]. When this approach is coupled with sensitive liquid chromatography-tandem mass spectrometry (LC-MS/MS), proteins that are enriched on mitotic chromosomes en masse[40], or individual chromosomes, can be quantitatively identified. Here, to identify factors that might convey memory of Xi/Xa chromatin states during cell division, we isolate mitotic Xa and Xi chromosomes directly from female mouse B-lymphocyte progenitors (mpre-B) by high-resolution flow cytometry and perform proteomics on individual chromosomes using LC-MS/MS. A biphasic distribution of H3K27me3 on X chromosomes in female cells allows us to separately purify Xi and Xa metaphase chromosomes and compare their size, chromatin structure, and proteomes. Unexpectedly, our analysis reveals a significant enrichment (or retention) of components of Hbo1 and Mof-Msl histone acetyltransferase (HAT) complexes on the Xa as compared to Xi or autosomes. While differences in histone acetylation between Xi and Xa are known in interphase, differences in mitosis are unexpected as mitotic chromosome condensation, phase transition, and chromosome segregation require global deacetylation[67–72]. Here, we show that removal of components of Hbo1 or Mof-Msl complexes compromises differences in chromosome compaction and H3K27me3 distribution of Xi and Xa metaphase chromosomes, revealing an unanticipated role for histone acetylation in enforcing the mitotic status of the active X chromosome.

## Results

### Xist-RNA dissociates from the Xi at anaphase in mpre-B cells

To examine Xist distribution through cell division, male and female mpre-B cells were isolated from foetal liver and immortalized with Abelson murine leukaemia virus, as described previously[73]. Male cells contained a single X chromosome as revealed by DNA fluorescence in situ hybridization (FISH) analysis, whereas two X-chromosome-specific signals were detected in female cells (green, Fig. 1a left panels). As expected, Xist-RNA was uniquely seen in female cells (red, Fig. 1a middle panels) where signal focused around the presumptive Xi with high levels of H3K27me3 at the DAPI-intense Barr body (Fig. 1a, right panels). During interphase, this pattern of Xist signal was evident in the majority of female mpre-B cells (>97%), while Xist-RNA was never detected in male cells (lower panels of Fig. 1a). The identity of male and female cells was confirmed by molecular detection of either a single X and Y chromosome or two X chromosomes, respectively (Extended Data Fig. 1a,b).

The distribution of Xist-RNA was also examined in cells undergoing division. Among asynchronously dividing cultures of female mpre-B cells, Xist-RNA was detected in most cells at prophase (99%), although the intensity of this signal was reduced as compared to interphase and further declined as chromosomes condensed (Fig. 1b,c). In metaphase

---

**Fig. 1 | Identifying proteins bound to native mitotic X chromosomes isolated from female and male mpre-B cells. a**, Representative images of female and male interphase mpre-B cells labelled with chromosome X paint (green, left), Xist RNA FISH probe (red, middle panel) or H3K27me3 antibody (red, right panel). DAPI stain is shown in blue. Scale bars, 4 μm. Images are representative of three independent experiments. **b**, Representative RNA FISH images of female and male mpre-B cells labelled for Xist (red) at different stages of mitosis (prophase, metaphase, anaphase and telophase). DAPI stain is shown in blue. Scale bars, 4 μm. Images are representative of three independent experiments. **c**, Table showing Xist RNA signal detected in female mpre-B cells at each stage of mitosis. Data collected over three independent experiments. **d**, Scheme of experimental strategy used to isolate native metaphase chromosome X and 19 from metaphase-arrested female and male mpre-B cells, and identify their chromosome-bound factors. Representative Hoechst 33258 and Chromomycin A3 bivariate karyotype (obtained by flow cytometry) of male mpre-B cells. Gates used to sort chromosome X and 19 are indicated. Proteomic analysis was performed on one million chromosomes (X or 19) using LC-MS/MS. **e**, Representative images of female (left) and male (right) mitotic chromosome FACS plots. Gates used to purify chromosome X and 19 are indicated. FACS plots are representative of six independent experiments. **f**, Representative images of flow-sorted mpre-B mitotic chromosomes 19 and X stained with mouse chromosome 19-specific (green) and X-specific (green) DNA FISH probes. Values indicate percentage of chromosomes labelled with each probe. The number of chromosomes analysed was 682 and 812 for chromosome X and 19, respectively. Scale bars, 2 μm. **g**, Volcano plots showing proteins significantly enriched on female (red) and male (blue) chromosome X (left) and chromosome 19 (right) isolated from mpre-B cell lines. After MaxQuant analysis, volcano plots were generated on Perseus using unpaired two-tailed Student's *t*-test, permutation-based FDR < 0.05, s0 = 0.1, *n* = six biological replicates (see Methods for detailed analysis). Proteins were plotted as $\log_2$(fold change) (LFQ intensity of flow-purified male over female mitotic chromosomes) versus significance ($-\log_{10}$(*P* value)), using Perseus software. Total protein hits, and the number of proteins enriched on female and male chromosomes X (left) and 19 (right) are indicated on the respective volcano plots. MacroH2A (H2afy) is highlighted in red. **h,i**, As in **g** but highlighting Xist RNA-associated factors (**h**) and components of repressive chromatin complexes (**i**).

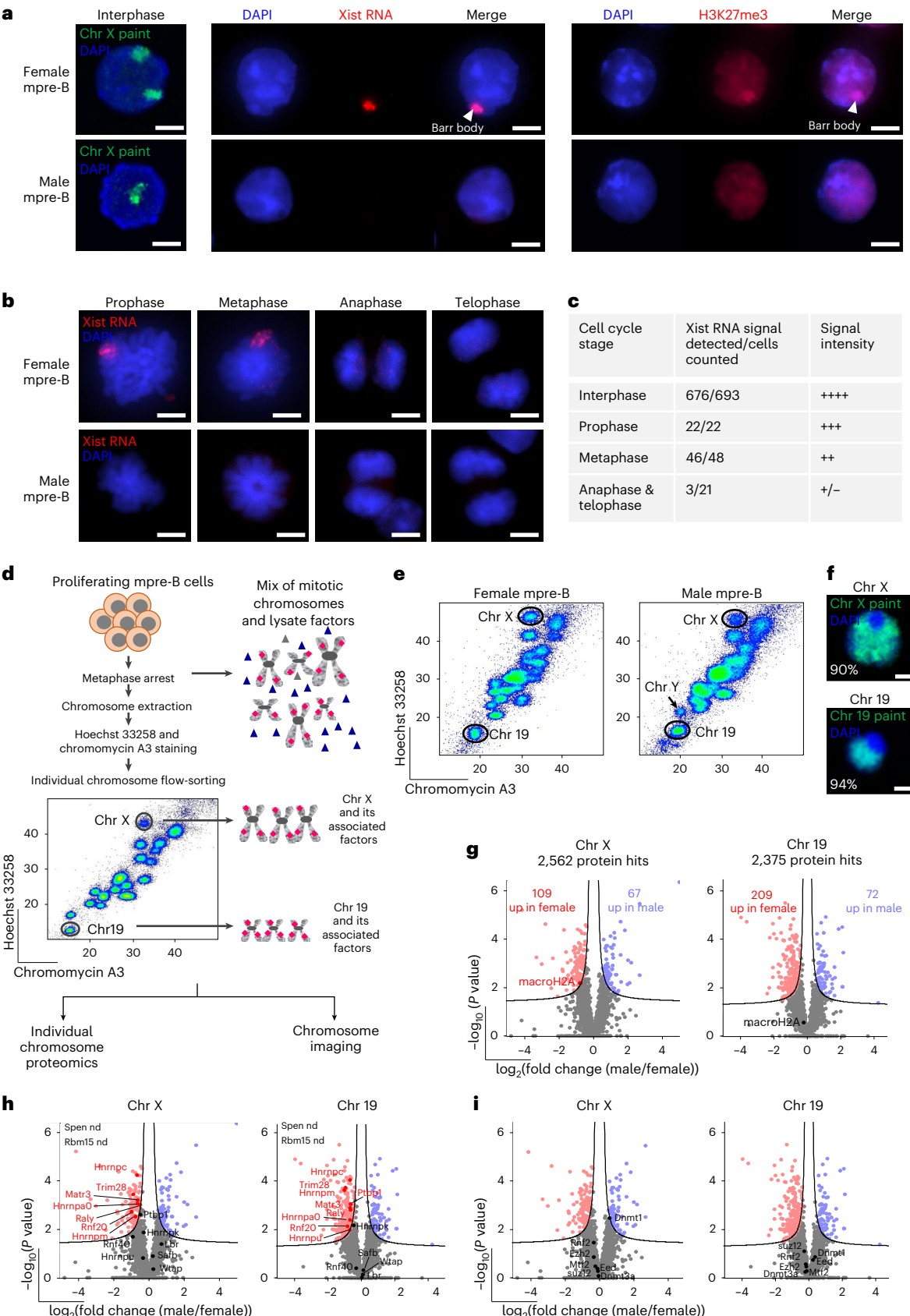

Xist-RNA signal was still evident at the presumptive Xi, but this was lost as cells entered anaphase and was undetectable at telophase. These data reaffirm the known displacement of Xist-RNA from chromosomes during mitosis[74–76].

## Proteomic comparison of mitotic chromosomes from male or female mpre-B cells

To investigate potential factors that could convey epigenetic memory of X chromosome states during cell division, we isolated individual X chromosomes from male and female mpre-B cells and performed proteomic analysis. These cells were derived from littermates and have stable karyotypes and Xist-RNA expression. The rationale for this analysis is that in female cells, 50% of X chromosomes are active and 50% inactive, whereas in male cells (XY), all X chromosomes are active. Consequently, differences between female and male chromosome X should be attributable to Xi. To identify factors uniquely enriched on Xi, we performed proteomics on X chromosomes isolated from metaphase-arrested male and female mpre-B cells (Fig. 1d) using mouse chromosome 19 as an autosomal comparator (Fig. 1d,e). As previously described[40], chromosome suspensions were prepared from male and female metaphase-arrested cells, stained with DNA dyes Hoechst 33258 and Chromomycin A3 to enable visualization and isolation by flow cytometry of mitotic chromosomes en masse, or individually (according to GC:AT ratio) (Fig. 1e). Purity of isolated chromosomes X and 19 samples was verified using DNA FISH and chromosome-specific probes and estimated to be 90% and 94%, respectively (Fig. 1f).

One million individual flow-sorted chromosomes X and 19 were subjected to quantitative LC-MS/MS proteomics. Principal component analysis (PCA) showed that female-derived and male-derived samples clustered separately for both X and 19 (Extended Data Fig. 1c,d). A total of 2,562 protein hits were detected in female versus male chromosomes X samples, and 2,375 proteins hits in female versus male chromosome 19 samples (Supplementary Tables 1 and 2). Of these, 109 showed a significant enrichment on female X chromosomes as compared to male (Fig. 1g) including histone gene macroH2A (H2afy), depicted in red, a histone variant that is preferentially incorporated at the Xi in interphase (Extended Data Fig. 1e) and maintained during cell division[60,77].

Among the proteins enriched on X chromosomes isolated from female versus male samples, we identified several candidates reported to associate with Xist-RNA[39,53,78,79] (Fig. 1h left, highlighted

in red). However, as these candidates showed a similar enrichment on female chromosome 19 (Fig. 1h right), preferential retention of these factors was not specific for the X and most likely reflects slight differences in protein abundance between the two cell lines. In addition, several chromatin repressor complexes that remain chromosome-associated throughout mitosis, such as DNMT1,3a, PRC2 and PRC1 (refs. 40,66,80), showed no increased representation in female samples (that contain Xi) compared to male (Fig. 1i).

In view of these results, we asked whether Xist-RNA was present in the chromosome samples used for proteomic analyses. This is important because our results had shown that Xist-RNA was precipitously lost at the transition from metaphase to anaphase (Fig. 1b,c). RNA-FISH analysis confirmed that Xist-RNA coats the presumptive Xi in non-arrested mpre-B cells at metaphase (Extended Data Fig. 1f). However, in cultures treated with demecolcine to induce 'metaphase-arrest', Xist-RNA signal was detected in only a minority of cells (1 in 20) (middle panel, Extended Data Fig. 1f). This suggests that although demecolcine acts to prevent mitotic spindle formation[81], it probably does not prevent the ongoing displacement of Xist from the Xi. Consistent with this, Xist-RNA was detected in a minority (1/40) of flow-sorted X chromosomes isolated from demecolcine-treated female mpre-Bs (right panel, Extended Data Fig. 1f). In these samples Xist signal was punctate, reminiscent of Xist particles that have been reported by others[82,83]. Taken together, these results confirm Xist displacement from Xi as cells transit mitosis.

A comprehensive comparison of proteins enriched on female versus male X chromosomes (Extended Data Fig. 1g,h) revealed only a relatively small number of candidates (37 and 33) that were enriched solely on female or male X chromosomes, with no equivalent enrichment on female or male chromosome 19 (lists provided in Supplementary Table 3).

## Mitotic Xa and Xi isolated based on H3K27me3 show differential compaction

Since female samples contain both Xi and Xa chromosomes, it is possible that heterogeneity acts as a confounder in the detection of proteomic differences between mitotic Xi and Xa chromosomes. To resolve this, we adopted an alternative approach where elevated levels of H3K27me3 at Xi was used to distinguish Xa from Xi metaphase chromosomes by flow cytometry. Mitotic chromosome preparation was modified by adding one additional step consisting of staining

**Fig. 2 | Flow-sorting of mitotic Xa and Xi and identification of chromosome-associated factors. a**, Outline of the protocol used to flow-purify mitotic Xa and Xi directly from female mpre-B cells. **b**, Gating used to flow-isolate mitotic Xi and Xa from female mpre-B cells based on H3K27me3 staining. Histograms showing H3K27me3 staining profile of female mitotic X chromosomes (left panels) and male mitotic X chromosomes (right panels). **c**, Bisulfite analysis of CpG DNA methylation on H3K27me3 high and H3K27me3 low X chromosomes. Diagram (top) shows the CGIs analysed by bisulfite PCR. The black line shows the location of the region analysed relative to the gene transcription start site (arrows) for *Hprt* (left panels) and *Pdk3* (right panels) genes. Each circle represents a CpG dinucleotide and each line represents methylation on an individual DNA strand determined by sequencing subcloned PCR product from bisulfite-treated genomic DNA. Black circles, methylated CpGs; white circles, unmethylated CpGs. Percentage of methylation is indicated in brackets. **d**, Immunofluorescence labelling of macroH2a (green) on flow-sorted mitotic female Xi, female Xa and male X chromosomes isolated from female and male mpre-B cells. DAPI counterstain is shown in light grey. Scale bars, 2 µm. Images are representative of three independent experiments showing consistently higher level of macroH2A on Xi compared to Xa. **e**, Representative images of immunolabelling of H3K27me3 (red) on flow-sorted mitotic female Xi, female Xa and male X chromosomes isolated from female and male mpre-B cells. DAPI counterstain is shown in light grey. Scale bars, 2 µm. Plots (right of the images) show H3K27me3 mean intensity quantification for each chromosome (lower graph) and the corresponding chromosome size measurements (upper graph). **f**, Representative images of immunolabelling of H3K27me3 (red) on flow-sorted

mitotic female and male chromosome 19 isolated from female and male mpre-B cells. DAPI counterstain is shown in light grey. Scale bars, 2 µm. Histograms (right of the images) show H3K27me3 mean intensity quantification for each male and female chromosome 19 (lower graph) and the corresponding chromosome size measurements (Upper graph). For H3K27me3 intensity plots in **e** and **f**: mean ± SD are shown. *P* values of statistically significant changes, measured by unpaired two-tailed Student's *t*-tests, are indicated. For chromosome size measurements: minimum, lower quartile, median, upper quartile and maximum values are indicated. *P* values of statistically significant changes, measured by unpaired two-tailed Student's *t*-tests, are indicated. $P = 2.8 \times 10^{-41}$ and $P = 5 \times 10^{-22}$ (**e**, upper plot), $P = 6 \times 10^{-50}$ and $P = 3 \times 10^{-39}$ (**e**, lower plot); ns, not significant (**f**). Graphs are representative of three independent experiments. Number of chromosomes analysed: 102, 100 and 101 for female Xi, Xa and male X chromosomes, respectively; 122 and 119 for female and male chromosomes 19, respectively. Data collected over three independent experiments. **g**, Volcano plots of proteins significantly enriched on female Xi (red) and Xa (green) isolated from mpre-B cell lines. Unpaired two-tailed Student's *t*-test, permutation-based FDR < 0.05, s0 = 0.1, *n* = four independent replicates for Xi and three for Xa. Proteins were plotted as log₂(fold change) (LFQ intensity of flow-purified mitotic Xa over Xi chromosome) versus significance (−log₁₀(*P*)), using Perseus software. Total protein hits, and the number of proteins enriched on Xi and Xa are indicated on the volcano plots. MacroH2A is highlighted. **h,i**, As in **g** but highlighting components of repressive chromatin complexes (**h**) and Xist RNA-associated factors (**i**). nd, not detected. Source data are available for **e** and **f**.

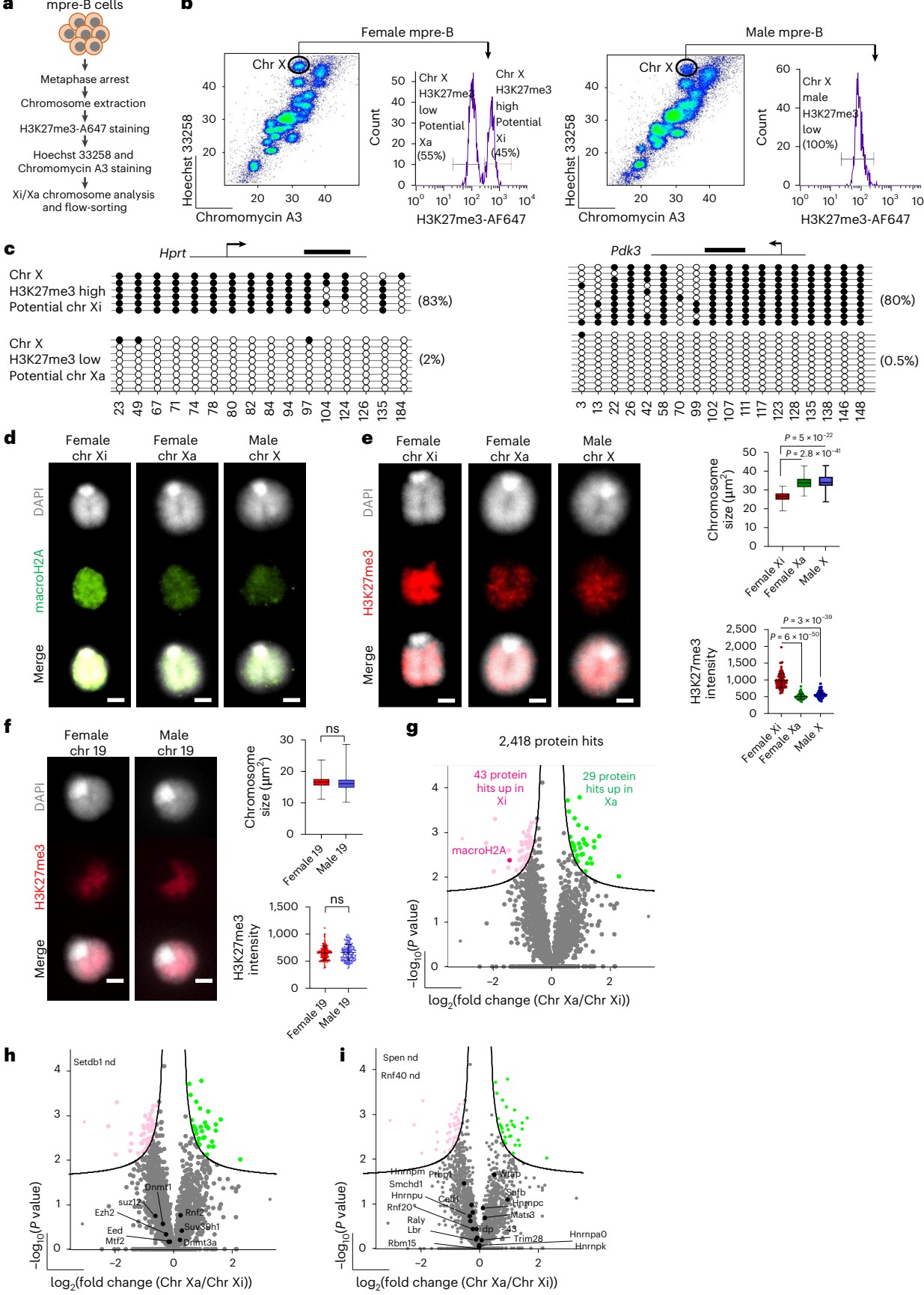

with a directly coupled antibody (H3K27me3-AF647) that recognises trimethylated H3K27, prior to staining with Hoechst 33258 and Chromomycin A3 (outlined in Fig. 2a). This enables two populations (high- and low-H3K27me3) to be identified among female X chromosomes. As shown in Fig. 2b, H3K27me3 distribution was bimodal in female mpre-B cells (left histogram), whereas H3K27me3 distribution was unimodal in male X chromosomes (right histogram). Native mitotic chromosomes corresponding to the presumptive Xi (high-H3K27me3) and Xa (low-H3K27me3) chromosomes were separately isolated by flow cytometry. Parallel analyses of two autosomes derived from female cells (chromosomes 3 and 19) confirmed that bimodal H3K27me3 distribution was a unique feature of female X chromosomes (Extended Data Fig. 2a). To validate the epigenetic status of purified X chromosomes, we analysed DNA methylation at selected CpG islands of X linked genes. In the high-H3K27me3 population we observed extensive CpG methylation at two X-linked genes *Hprt* and *Pdk3* (Fig. 2c) while the *Xist* locus was unmethylated (Extended Data Fig. 2b). Conversely, in the low-H3K27me3 chromosome X population, *Hprt* and *Pdk3* were unmethylated, while *Xist* showed extensive CpG methylation. *Eif2s3x*, an X-linked locus that escape inactivation[84], was shown to be similarly unmethylated in high- and low-H3K27me3 samples, as anticipated (Extended Data Fig. 2b, right panel). These results confirmed that female X chromosomes isolated as high-H3K27me3 represent the Xi, while those isolated as low-H3K27me3 comprise the Xa.

To investigate whether Xi and Xa display different structures or sizes at metaphase, DNA FISH, immunolabelling and microscopy was used to examine isolated Xi and Xa chromosomes. As predicted, metaphase Xi and Xa chromosomes were differentially labelled with antibody to macroH2A (Fig. 2d and quantified in Extended Data Fig. 2c). As anticipated, H3K27me3 labelling was more intense at Xi than Xa or male X chromosomes (Fig. 2e, quantified in lower dot plot). In addition, Xi chromosomes were significantly smaller and more compact at metaphase than their active counterparts, whether isolated from female or male cells (Fig. 2e, and upper graph). In comparison, chromosome 19 showed a similar H3K27me3 intensity and size in samples derived from male or female mpre-B cells (Fig. 2f). Taken together these results indicate that some of the distinctive differences between the Xa and Xi, such as chromatin state and relative compaction, previously reported in interphase[31-33], are retained as chromosomes condense at metaphase.

**Proteomic comparison of Xa and Xi metaphase chromosomes**
Proteomic comparisons were performed on an equivalent number (one million) of metaphase Xa, Xi chromosomes from females, as well as male X chromosomes and revealed that of 2,418 candidate proteins identified, 29 showed significant enrichment on the Xa (as compared to the Xi), and 43 were significantly enriched on the Xi (Fig. 2g, Supplementary Tables 4 and 5). PCA comparing Xa, Xi and male X chromosome samples, showed that each sample category clusters separately (Extended Data Fig. 2d), and as anticipated macroH2A showed a significant enrichment selectively on Xi (Fig. 2g, highlighted in red). No significant difference of macroH2A abundance was observed between female Xa and male X (Extended Data Fig. 2e). Although chromatin repressors co-enriched with mitotic chromosomes as compared to mitotic lysates (Extended Data Fig. 2f and Supplementary Table 6) consistent with other reports[40,66,80], we did not observe any enrichment of repressive complexes (such as DNMTs, PRC1, PRC2) on mitotic Xi as compared to Xa (Fig. 2h). We also did not detect any significant enrichment of Xist-associated proteins on mitotic Xi as compared to Xa (Fig. 2i), including Trim28 (Extended Data Fig. 2g), a B-cell specific Xist-interacting factor that has recently been implicated in XCI maintenance[85]. While acknowledging that there is still conflicting literature about Xist-associated proteins and XCI maintenance[6,27,85-88], we did not detect evidence of Xist-interacting proteins differentially marking the Xi in mitosis.

**HATs and acetyl-binding proteins are selectively enriched on mitotic Xa**
Several studies have reported that histone acetylation is globally reduced as cells enter mitosis, while histone methylation is preserved, and H3S10p increases[67,68,76,89]. In addition, histone deacetylase activity is important for chromosome compaction and segregation at cell division[68-72]. Against this backdrop we asked whether histone acetylation differed between mitotic Xa and Xi chromosomes, as reported in immunofluorescence based-studies[90,91]. We noted that subunits of several HAT complexes (Fig. 3a), acetyl-binding proteins (Fig. 3b) and proteins associated with active transcription (Fig. 3c), showed an enrichment on Xa as compared to Xi (upper panels, highlighted in green). This included Kat7, a component of Hbo1 HAT complex, as well as Msl1 (component of Msl complex), Morf4l1 and Ing5. Interestingly, as Kat7 and Msl1 also showed significant enrichment on total mitotic chromosomes as compared to equivalent mitotic cell lysates (Extended Data Fig. 3a), our results confirm that both these proteins are retained on the mitotic chromosomes of female mpre-B cells. Selective enrichment of Kat7 on the Xa versus Xi was also independently verified by quantitative immunofluorescent labelling of flow-sorted metaphase chromosomes, as shown in Extended Data Fig. 3b. Several histone acetyl-binding proteins and factors associated with active transcription were similarly enriched on female Xa relative to Xi (Fig. 3b,c, top panels). For example, Atad2b, an acetyl-binding protein and several proteins associated with active transcription were significantly and selectively co-enriched on female-derived Xa samples, versus female-derived Xi samples. Subunits of the MLL H3K4 methyltransferase and SWI/SNF remodelling complexes (Psip1, Smarcd1) and the Phf8 demethylase were also significantly enriched on Xa (Fig. 3c). Importantly, control comparisons made between mitotic Xa and male (active) X chromosomes (Fig. 3a-c, lower panels) showed no significant difference between these samples, except for Msl1. These data indicate that active X chromosomes derived from either male or female mitotic cells have a propensity to retain HAT complexes relative to Xi.

Comparing LFQ intensities of different subunits of HAT complexes (Hbo1, Moz/Morf and Msl) between mitotic Xa, Xi and chromosome 19 samples (Fig. 3d, left), suggests that Meaf6, Kat7 and Msl1 were all significantly enriched on the Xa as compared to Xi, or an autosomal control (chromosome 19). The abundance of macroH2A was higher on the Xi than all other chromosomes analysed (Fig. 3d, right). Since Hbo1, Moz/Morf and Msl complexes target histone H3 (at lysine 9, 14 and 23) and histone H4 (at lysine 5, 8, 12 and 16), we compared the levels of acetylated peptides in our proteomic dataset (Supplementary Table 7). We detected increases in the levels of several acetylated histone peptides in Xa versus Xi metaphase chromosome samples (Fig. 3e). In comparison, acetylated H2B (K5, K16) was unchanged between the Xa and Xi and importantly, total histone quantification of H2B, H3 and H4 was also comparable (Fig. 3e, right). Taken together, these results highlight significant differences in histone H3 and H4 acetylation levels between Xa and Xi mitotic chromosomes.

Among factors that showed a significant enrichment on the mitotic Xi relative to the Xa, we detected a non-canonical (histone deacetylase (HDAC)-independent) inhibitor of histone acetyltransferase activity, Noc2l (ref. 92; Extended Data Fig. 3c; validated by immunostaining, Extended Data Fig. 3d), and a sequence-specific DNA-binding protein, Mybbp1a, that functions as a repressor in part through HDAC activity[93,94] (Extended Data Fig. 3c). Noc2l and Mybbp1 were also more abundant on the Xi than other chromosomes analysed (Extended Data Fig. 3e). Collectively, these data show that differential histone acetylation pathways are a feature of Xi and Xa chromosomes in mitosis.

**Differential size of mitotic Xa versus Xi depends on histone H3K14ac**
To verify the histone acetylation modifications that distinguish X chromosomes in mitosis, we performed immunolabelling using

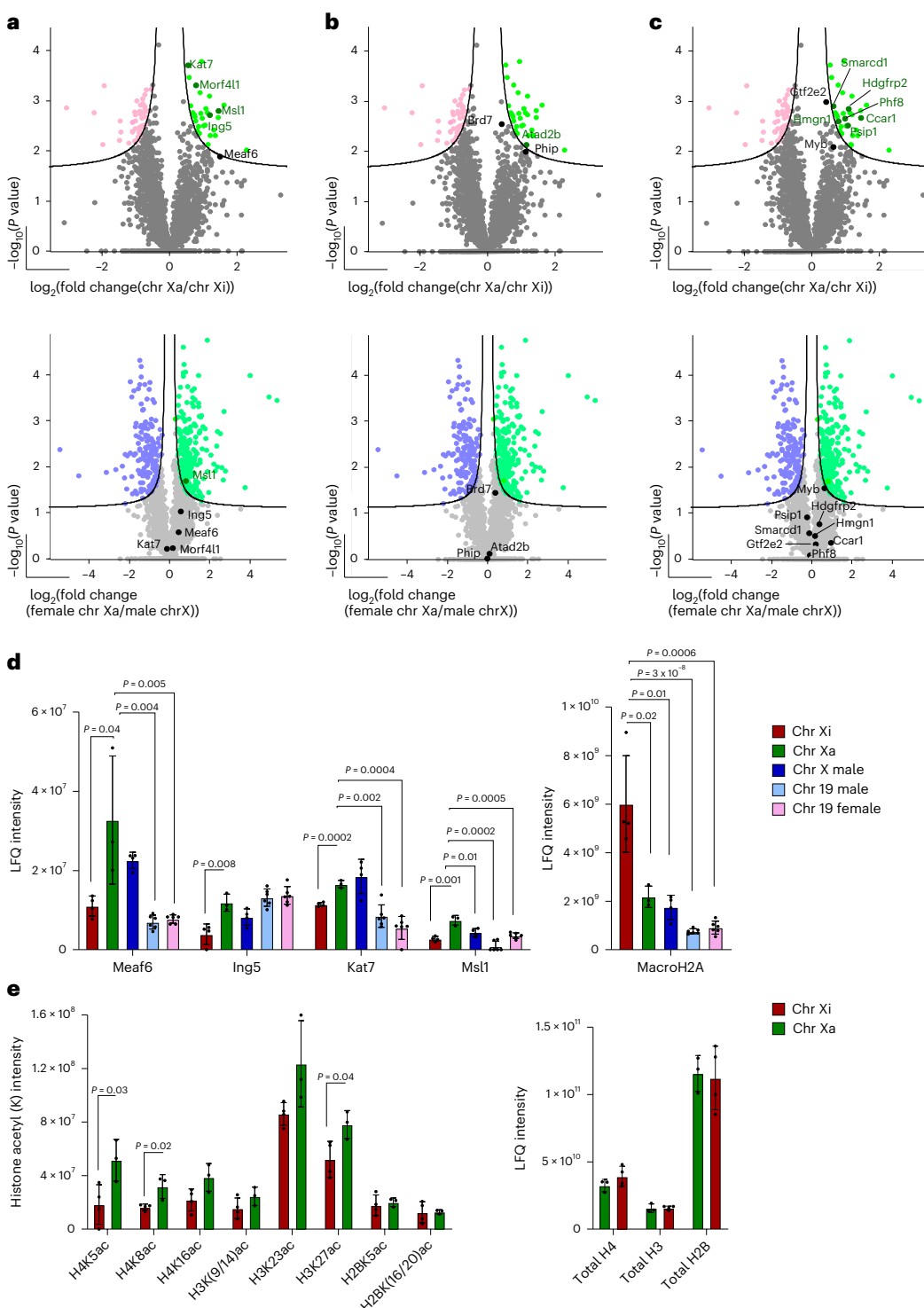

**Fig. 3 | Selective enrichment of HAT complexes and acetyl-binding proteins on mitotic Xa chromosomes. a–c,** Volcano plots of proteins significantly enriched on female Xi (red) versus female Xa (green) (upper panel) or on female Xa versus male X chromosomes (lower panel), isolated from a female and a male mpre-B cell line. Unpaired two-tailed Student's *t*-test, permutation-based FDR < 0.05, s0 = 0.1, *n* = four biological replicates for male X, female Xi and three for female Xa. Proteins were plotted as log$_2$(fold change) versus −log$_{10}$(*P*), using Perseus software. Components of HAT complexes (**a**) acetyl-binding proteins (**b**) and proteins associated with active transcription (**c**) are highlighted in the volcano plots. **d**, Left, average LFQ intensity of Meaf6, Ing5, Kat7 and Msl1 in female chromosome Xi, Xa, male chromosome X, and female chromosome 19 samples. Mean ± SD is shown, *n* = four biological replicates for male X and female Xi, three for female Xa, and six replicates for female and male chromosome 19.

*P* value of statistically significant changes relative to Xa, measured by unpaired two-tailed Student's *t*-tests, is indicated. Right, average LFQ intensity of macroH2A as in the left panel. *P* value of statistically significant changes relative to Xi, measured by unpaired two-tailed Student's *t*-tests, is indicated. *P* = 0.04, *P* = 0.004 and *P* = 0.005 (Meaf6). *P* = 0.008 (Ing5). *P* = 0.0002, *P* = 0.002 and *P* = 0.0004 (Kat7). *P* = 0.001, *P* = 0.01, *P* = 0.0002 and *P* = 0.0005 (Msl1). *P* = 0.02, p = 0.01, *P* = 3 × 10$^{-8}$ and *P* = 0.0006 (macroH2A). **e**, Average site intensity of histone acetylation modifications analysed from the same proteomic dataset as in **a** with additional filtering on Perseus (see details in Methods) (left panel). Average LFQ intensity of total histone H4, H3 and H2B (right panel). Mean ± SD is shown. *P* value of statistically significant change, measured by unpaired two-tailed Student's *t*-tests, is indicated. *P* = 0.03 (H4K5ac), *P* = 0.02 (H4K8ac), *P* = 0.04 (H3K27ac). Source data are available for **d** and **e**.

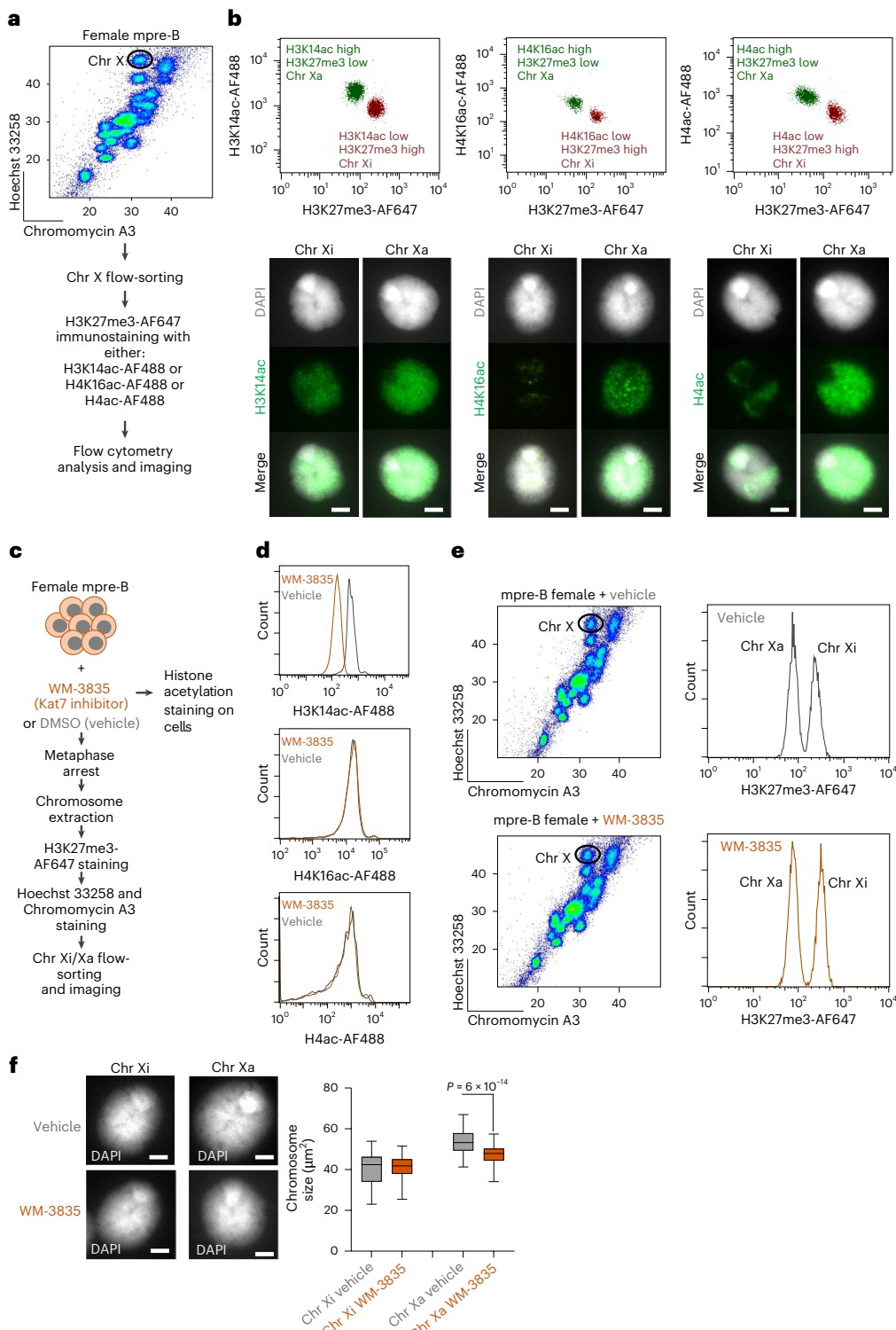

**Fig. 4 | Inhibiting Kat7 activity selectively reduces histone H3K14ac levels and increases compaction of mitotic Xa. a**, Experimental strategy used to analyse histone acetylation modifications on mitotic Xa and Xi using flow cytometry. **b**, Analysis of H3K14ac (left panel), H4K16ac (middle panel) and pan-acetyl H4 (H4K5 + K12 + K16ac) (right panel) labelling intensity on female mitotic Xi (H3K27me3 high) and Xa (H3K27me3 low) (top panels). Immunofluorescence images are shown in the lower panels. FACS plots and images are representative of three independent experiments. Scale bars, 2 μm. **c**, Outline of the experimental design used to analyse mitotic Xi and Xa following inhibition of Kat7/Hbo1 activity. **d**, Histone acetylation staining and flow cytometric analysis (H3K14ac top panel, H4K16ac middle panel, and pan-acetyl H4 lower panel) on female mpre-B cells treated with DMSO (vehicle) or WM-3835 (Kat7 inhibitor).

Histograms shown are representative of three independent experiments. **e**, H3K27me3 labelling intensity on X chromosomes derived from female mpre-B cells treated with DMSO or WM-3835. **f**, Representative images of flow-sorted mitotic Xa and Xi isolated from female mpre-B cells treated with DMSO or WM-3835, stained with DAPI (light grey). Scale bars, 2 μm. Plot (right of the images) show corresponding chromosome size measurements for Xi and Xa in each condition. Minimum, lower quartile, median, upper quartile and maximum values are indicated. Number of chromosomes analysed: 103, 112, 85 and 107 for Xi vehicle, Xi WM-3835, Xa vehicle and Xa WM-3835 chromosomes, respectively. Data collected over three independent experiments. $P$ values of statistically significant changes, measured by unpaired two-tailed Student's $t$-tests, are indicated. $P = 6 \times 10^{-14}$ (right plot). Source data are available for **f**.

antibodies that recognise pan-acetyl H4, acetyl H3K14 and acetyl H4K16, of female Xa and Xi metaphase chromosomes, isolated by flow cytometry (Fig. 4a). Figure 4b shows that Xa chromosomes could be defined as being low-H3K27me3 and high-acetyl H3/H4, relative to Xi chromosomes. Bimodal distribution of acetyl H4K16 (H4K16ac), acetyl H3K14 (H3K14ac) and pan-H4 acetylation (H4ac), displayed by mitotic X chromosomes, was not seen in female-derived autosomes such as chromosomes 3 and 19 (Extended Data Fig. 4a,b). To investigate the functional relevance of enhanced H3K14 acetylation on Xa, we treated female mpre-B cells with an Hbo1/Kat7-specific inhibitor, WM-3835 (ref. 95; Fig. 4c). Cells exposed to WM-3835 for 48 hours showed a global reduction in histone H3K14ac levels as compared to control cells (vehicle), while acetylation of H4K16 and pan acetylation of H4 were unaffected (Fig. 4d). WM-3835 treatment also provoked a decline in H3K14ac detected on isolated mitotic chromosomes X, 3 and 19 (Extended Data Fig. 4c).

To discover the impact of reduced H3K14ac on chromosomes 3, 19, Xi and Xa we isolated and studied native metaphase chromosomes from WM-3835-treated and vehicle-treated cells (Fig. 4e, and Extended Data Fig. 4d). Although WM-3835 treatment did not significantly affect H3K27me3 distribution on any chromosome (Fig. 4e and Extended Data Fig. 4d), and did not affect the size of the Xi (Fig. 4f), we observed a significant reduction in size of metaphase Xa chromosomes in response to experimentally induced H3K14ac reduction (Fig. 4f). The impact of WM-3835 on autosome size was variable (Extended Data Fig. 4e). Xist-RNA expression in interphase mpre-B cells did not change following Hbo1 inhibition (Extended Data Fig. 4f). These results show that loss of Hbo1-mediated H3K14ac enhanced Xa chromosome compaction, so that it more closely resembles the size of the Xi in mitosis.

## Mof-Msl sustain differential size and H3K27me3 profiles of mitotic Xi and Xa

Our results indicate that components of Mof/Msl complex and histone H4K16 acetylation were enriched in female Xa versus Xi metaphase chromosomes. To investigate the role of Mof/Msl-mediated H4K16 acetylation, mouse embryonic fibroblasts (MEFs) were derived from E13.5 embryos *Mof*^fl/fl *Cre-ERT2*^T/+ inducible knockout (KO) and Msl2 constitutive KO. mpre-B cell lines could not be derived from either Msl2 KO embryos or following induction of Mof knockdown suggesting a possible role for Mof/Msl complex in B cell development or survival. As we have previously shown that native metaphase chromosomes can be isolated from a range of cell types including fibroblasts[40], we purified chromosomes X, 3 and 19 from Msl2 knockout MEFs. As anticipated, female Msl2 KO MEFs showed reduced H4K16ac labelling relative to WT controls (Extended Data Fig. 5a,b). Msl2 KO and WT female MEFs

were arrested in metaphase and released chromosomes were stained with Hoechst 33258 and Chromomycin A3 (Fig. 5a). X chromosomes were purified by gating (as indicated) and stained with antibody to H3K27me3 (Fig. 5b). In X chromosomes of WT MEFs, a clear bimodal distribution for H3K27me3 was observed. In Msl2 KO X chromosome samples, however, bimodal H3K27me3 distribution was lost in favour of a distribution that clustered around intermediate values (Fig. 5b). Interestingly, this change in H3K27me3 levels was also accompanied by a significant reduction in X chromosome size (Fig. 5c). No or very little changes of H3K27me3, and variable chromosome size changes was observed for mitotic autosomes 3 and 19 (Extended Data Fig. 5c,d).

To strengthen evidence of a causal link between reduced H4K16ac and alterations in mitotic Xa chromatin, we also purified chromosomes from female MEFs *Mof*^fl/fl *Cre-ERT2*^T/+ (Fig. 5d and Extended Data Fig. 6a) in which Mof activity could be conditionally deleted upon the addition of tamoxifen (4-OHT). As anticipated, 4-OHT treatment resulted in a dramatic global reduction in Mof expression in MEFs *Mof*^fl/fl *Cre-ERT2*^T/+ cells (Extended Data Fig. 6b), and a significant decline in H4K16ac mean intensity (Extended Data Fig. 6c). Flow-isolated chromosomes X, 3 and 19 were prepared from metaphase- arrested *Mof*^fl/fl *Cre-ERT2*^T/+ cells treated with 4-OHT (or vehicle) (Fig. 5d, e and Extended Data Fig. 6d), and purity of X chromosome sorting was verified by chromosome paint DNA FISH as being >80% (Fig. 5f). Conditional depletion of Mof resulted in loss of differential H3K27me3 distribution on X chromosomes, with a shift in the profile from bimodal (in vehicle treated) to unimodal in 4-OHT induced samples (Fig. 5e histograms, right). In addition, conditional Mof deletion and reduced histone H4K16ac resulted in reduced metaphase X chromosome size (Fig. 5g), with modest changes to chromosomes 3 and 19 (Extended Data Fig. 6d,e). Interestingly, metaphase X chromosomes isolated from female 4-OHT treated cells were much more compact than vehicle-treated Xa or Xi controls, suggesting that Mof depletion affects the behaviours of Xa and Xi chromosomes. Loss of Mof activity clearly has a global impact on chromosome/chromatin state and H4K16ac loss affects levels of H3K27me3 on autosomes as well as on X chromosomes. However, to formally exclude that 4-OHT treatment (and consequent loss of Mof activity) had resulted in X-chromosome loss, rather than altering the properties of Xi and Xa chromosomes, we performed additional DNA FISH analysis using X-paints, and examined Xist-RNA expression, in MEFs treated with 4-OHT or vehicle (Fig. 5h). Xist-RNA distribution was not significantly affected in Mof KO fibroblasts (Fig. 5i) and most fibroblasts retained a similar number of X chromosomes in both 4-OHT or vehicle conditions. This excludes that loss of bimodal H3K27me3 distribution was the result of a selective loss of Xi in Mof-deficient cells.

**Fig. 5 | Mof and Msl are required to sustain differential compaction and H3K27me3 profiles of X chromosomes in female fibroblasts. a**, Experimental procedure for isolating mitotic chromosomes from WT and Msl2 KO female MEFs. **b**, Hoechst 33258 and Chromomycin A3 bivariate mitotic chromosome flow karyotype (left panels) of WT and Msl2 KO female MEFs. Gates used to sort chromosome X are indicated. H3K27me3 staining profile (right panels) of female mitotic X chromosomes from WT and Msl2 KO female MEFs. **c**, Representative images of flow-sorted X chromosomes from WT and Msl2 KO female MEFs (left panel). DAPI counterstain is shown in light grey. Scale bars, 2 μm. Chromosome size measurements are shown in the right plot. Minimum, lower quartile, median, upper quartile and maximum values are indicated. Number of chromosomes analysed: 34, 47 and 49 for Xi WT, Xa WT and Msl2 KO X chromosomes, respectively. Data collected over three independent experiments. *P* values of statistically significant decreases, measured by unpaired two-tailed *t*-tests, are indicated. *P* = 3 × 10^−18 (right plot). **d**, Experimental procedure for isolating mitotic chromosomes from *Mof*^fl/fl *Cre-ERT2*^T/+ female MEFs treated with 3 days of 4-OHT or vehicle (EtOH). **e**, Flow karyotype (left panels) of mitotic chromosomes isolated from *Mof*^fl/fl *Cre-ERT2*^T/+ female MEFs treated with 3 days of 4-OHT or vehicle. Gates used to sort X chromosomes are indicated. Corresponding H3K27me3 intensity profiles are shown in the

right panels. **f**, DNA FISH analysis of mitotic X chromosomes isolated from *Mof*^fl/fl *Cre-ERT2*^T/+ female MEFs treated with 3 days of 4-OHT or vehicle, labelled with an X chromosome-specific paint. DAPI counterstain is shown in blue. Values indicate percentage of chromosomes labelled with each probe. Scale bars, 2 μm. Number of chromosomes analysed: 381 and 481 for chromosome X and chromosome X 4-OHT treated, respectively. Data collected over three independent experiments. **g**, Representative images (left) and size measurements (right) of X chromosomes isolated from *Mof*^fl/fl *Cre-ERT2*^T/+ female MEFs treated with 4-OHT or vehicle. DAPI counterstain is shown in light grey. Scale bars, 2 μm. For chromosome size measurements: minimum, lower quartile, median, upper quartile and maximum values are indicated. Number of chromosomes analysed: 25, 25 and 34 for Xi vehicle, Xa vehicle and X 4-OHT chromosomes, respectively. Data collected over three independent experiments. *P* values of statistically significant decreases, measured by unpaired two-tailed Student's t-test, are indicated. *P* = 6 × 10^−8 and *P* = 5 × 10^−16 (right plot). **h**, DNA FISH signals for chromosomes X and 3 detected in *Mof*^fl/fl *Cre-ERT2*^T/+ female MEFs treated with 4-OHT or vehicle. **i**, Xist RNA detection (red) in interphase *Mof*^fl/fl *Cre-ERT2*^T/+ female MEFs treated with 4-OHT or vehicle. DAPI stain is shown in blue. Scale bars, 4 μm. RNA-FISH images are representative of three independent experiments. Percentage of cells with one Xist RNA domain is indicated. Source data are available for **c** and **g**.

## Discussion

Here we show that many of the important differences in chromatin marking and chromosome condensation of active and inactive X chromosomes at interphase[29,33,96,97], are also conveyed through mitosis. These include increased DNA methylation, H3K27me3, macroH2A abundance and enhanced compaction of Xi, relative to Xa chromosomes. Some distinguishing features of the Xi, however, change as female mpreB cells enter mitosis. Xist-RNA dissociates from chromosomes post-metaphase, and we have shown that many proteins reported to

associate with XIST in human B cells[85] are displaced from Xi during mitosis. Our proteomic analyses did not detect Spen and Rbm15 components among metaphase lysates but indicated that although PRC2 (and some PRC1) components are retained on mitotic chromosomes[40,80], they show no significant enrichment on metaphase Xi, relative to Xa. This might be viewed as surprising as PRC1, Spen, Rbm15 and NCoR-Hdac3, are heavily implicated in initiating Xist-mediated silencing[50,52,53]. However, previous studies show that Xist-RNA is re-synthesized immediately after cytokinesis, dynamically re-coats Xi chromosomes early in G1

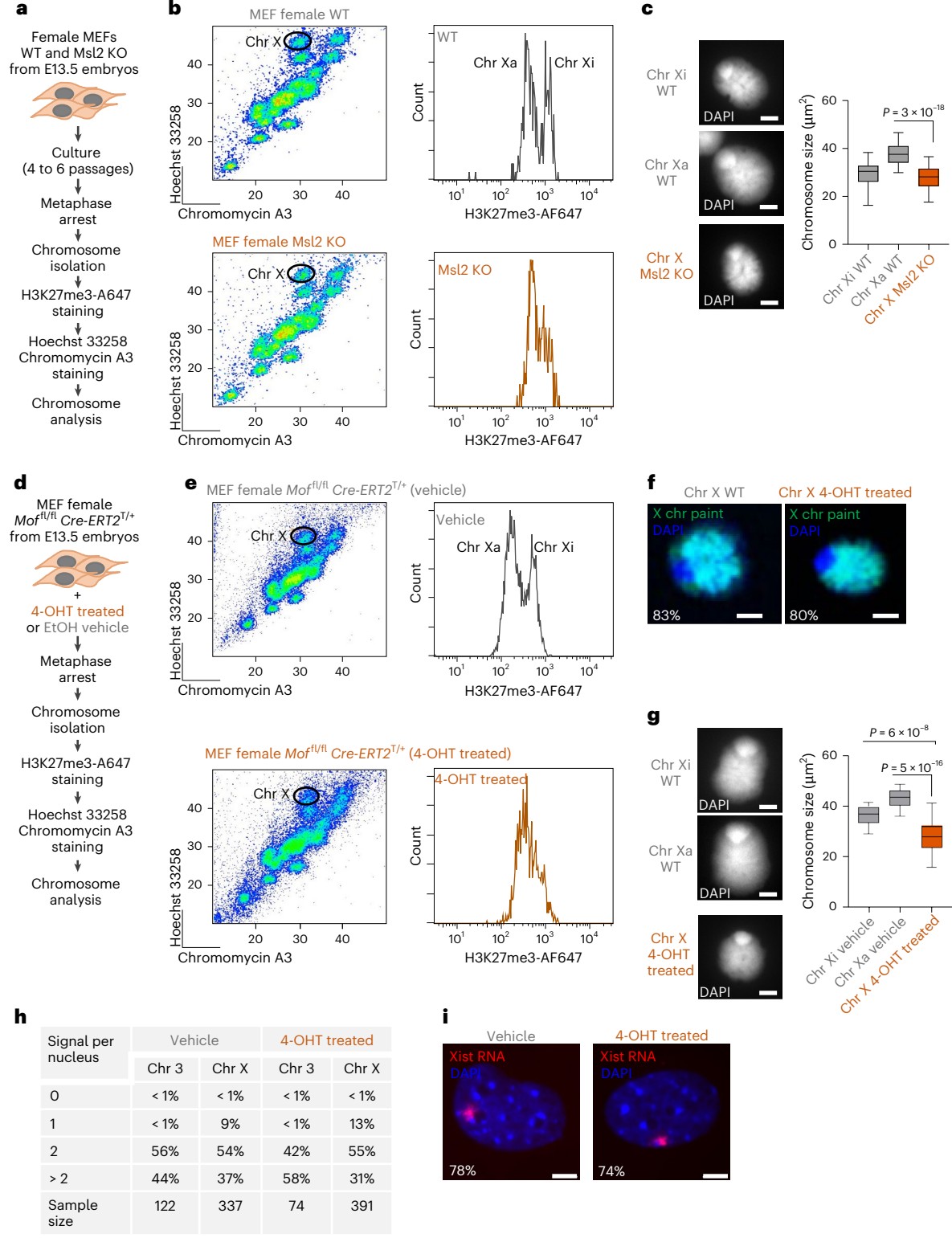

| Signal per nucleus | Vehicle | | 4-OHT treated | |
|---|---|---|---|---|
| | Chr 3 | Chr X | Chr 3 | Chr X |
| 0 | < 1% | < 1% | < 1% | < 1% |
| 1 | < 1% | 9% | < 1% | 13% |
| 2 | 56% | 54% | 42% | 55% |
| > 2 | 44% | 37% | 58% | 31% |
| Sample size | 122 | 337 | 74 | 391 |

phase[75,82]. Proteomic comparisons of mitotic mpre-B lysates and metaphase chromosomes (Extended Data Fig. 2f), showed that TRIM28, and other proteins implicated in a B-cell specific XIST complex[85], are evicted from metaphase chromosomes. Xist-associated proteins have been postulated to underpin Xist-mediated silencing, Xist-RNA stability, and spreading *in cis* along the chromosome during XCI initiation[39,53,58,78,79,98]. For these processes to operate at the beginning of each cell cycle, it seems likely that many Xist-associated proteins are simply recycled during mitosis via the activity of opposing kinase and phosphatases that mediate their eviction and re-instatement on chromosomes[99]. This prediction offers to meld current knowledge of Xist-RNA dynamics and resynthesis, with the proteomic data presented herein. The demonstration that SAF-A (hnRNPU), a the nuclear matrix protein important for Xist localisation on Xi[100,101], is evicted from chromosomes in prophase by Aurora kinase, and reinstated by PP1 (ref. 102), lends some support to such a proposal.

While early studies suggested that histone acetylation is maintained on metaphase chromosomes[90,103], more recent studies indicate a global histone deacetylation occurs and is required for entry into mitosis, chromosome condensation and segregation[68–72]. Here, we show that histone H3 and H4 acetylation is significantly elevated on mitotic female Xa, as compared to Xi, and components of several HAT complexes, including Hbo1/Kat7, Moz/Kat6b and Mof/Msl, are significantly enriched on Xa relative to Xi, or autosomal mean values. Hbo1 is a member of the Myst acetyltransferase family that forms complexes with different Jade or Brpf scaffold proteins to alternatively acetylate histone H3 (lysine 14 and 23), or histone H4 (lysine 5, 8 or 12), respectively[104]. We showed that Hbo1 inhibition, or deletion of Mof/Msl complex components, abrogates two distinguishing features of metaphase Xa and Xi chromosomes, their size and the relative abundance of histone H3K27me3. In female mpre-B cells, Hbo1/Kat7 inhibition increased compaction of metaphase Xa chromosomes (and autosome 19), without altering the sizes of Xi or a similarly sized autosome, chromosome 3. In fibroblasts, deletion of Mof/Msl complex components resulted in global decrease in H4K16ac and significantly alters mitotic chromosome size and H3K27me3 distribution, so that Xi and Xa could no longer be distinguished from each other based on these features.

As Xi is considered to be hypoacetylated[64,90,103,105], it could be argued that loss of HAT activity might preferentially affect the active (acetylated) X chromosome. However, as Mof deletion in female cells resulted in more compact X chromosomes (Fig. 5g) and increased in H3K27me3 levels on mitotic autosomes (Extended Data Fig. 6d), our data suggest that active and inactive X chromosomes are responsive to Mof/H4K16ac loss. Taken together these data indicate that key structural features that normally distinguish Xa chromosomes from Xi chromosomes, such as their relative size and repressive H3K27me3 abundance, require continuous HAT activity. In Drosophila, Msl2-mediated acetylation of H4K16 was shown to be instructive for dosage compensation[106–108]. In mammals, Msl2 was also shown to regulate allelic expression, and sustain the biallelic expression of gene cohorts that are normally expressed mono-allelically[109] by protecting alleles that are sensitive to silencing by DNA methylation. While we do not yet understand the mechanisms by which a loss of Mof/Msl2 activity changes the distribution of H3K27me3 on female X chromosomes, it is possible that Msl2 also protects the Xi from an erosion of repressive H3K27me3. Importantly, we have ruled out that altered H3K27me3 distribution is the result of X chromosome loss, or compromised Xist expression.

Our data reveal an unanticipated role of HAT complexes enriched on the Xa in maintaining differential X chromosome size and H3K27me3 marking. We identified two nucleolar-associated proteins, Noc2l and Mybbp1a, that are enriched on the Xi and inhibit histone acetylation through different mechanisms. Noc2l inhibits histone acetyltransferase, via a HDAC-independent mechanism[92,110], while Mybbp1a is a sequence-specific transcriptional repressor that functions in part

through HDAC activity[93,94]. It is possible that their enrichment on metaphase Xi chromosomes, represents a 'carry over' of proteins because of the juxtaposition of Xi chromosomes with the nuclear periphery and nucleoli in interphase. Alternatively, these factors may function to reinforce low-level histone acetylation on Xi chromosomes. In either case, HAT enrichment on mouse metaphase Xa chromosomes prompts a reconsideration of whether increased histone acetylation could be part of (or a vestigial product of), the need to balance gene dose between X chromosomes and autosomes. In Ohno's hypothesis, dosage compensation in mammals evolved in two discrete steps: hyper-activation of X chromosomes in both sexes (to compensate for gene losses on the Y), and random inactivation of one or other of the X chromosomes in female somatic cells (to balance dosage between sexes). Data supporting or refuting Ohno's proposal are difficult to interpret when based solely on measuring minute changes in gene expression, at near saturating levels. Our data, along with recent chromatin accessibility studies[111] offers an approach for tackling this dilemma, as sex chromosomes and autosomes can be directly purified from a range of different species and cell types and compared using quantitative proteomic approaches.

This study describes how individual mitotic chromosomes can be purified from metaphase cells, and their proteomes and properties interrogated using as few as $10^6$ sorted chromosomes. Active and inactive X chromosomes derived from female mpre-B cells, or primary fibroblasts, can be discriminated based on H3K27me3 and compared to each other, or to autosomes. In addition to HAT complexes, we uncovered a cohort of proteins enriched on Xa, relative to Xi, including Psip1, Smarcd1, Phf8, Cbx1 and Cbx3, that warrant investigation in future studies. We also provide evidence of increased representation of many RNA-binding proteins that are part of a B-cell specific XIST complex[85] on female-derived X and 19 metaphase chromosomes, compared with equivalents isolated from males. Although this result is intriguing and may relate to recent claims that Xist ribonucleoproteins promote female sex-biased autoimmunity[112], this awaits confirmation. Regardless of this caveat, our capacity to harness flow cytometry to isolate individual native mitotic chromosomes derived from different sexes, cell types or genetic backgrounds, provides a powerful biological template that is suitable for a variety of genetic and epigenetic experiments.

## Online content

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

[1]Department of Biochemistry, University of Oxford, Oxford, UK. [2]Epigenetic Memory Group, MRC Laboratory of Medical Sciences, Faculty of Medicine, Imperial College London, London, UK. [3]Flow Cytometry Facility, MRC Laboratory of Medical Sciences, Faculty of Medicine, Imperial College London, London, UK. [4]Mass spectrometry Facility, MRC Laboratory of Molecular Biology, Cambridge, UK. [5]Proteomics and Metabolomics Facility, MRC Laboratory of Medical Sciences, Faculty of Medicine, Imperial College London, London, UK. [6]Microscopy Facility, MRC Laboratory of Medical Sciences, Faculty of Medicine, Imperial College London, London, UK. [7]Max-Planck Institute of Immunobiology and Epigenetics, Freiburg, Germany. [8]Lymphocyte Development Group, MRC Laboratory of Medical Sciences, Faculty of Medicine, Imperial College London, London, UK. [9]These authors contributed equally: Dounia Djeghloul, Sherry Cheriyamkunnel. ✉e-mail: dounia-zede.djeghloul@cnrs.fr; amanda.fisher@bioch.ox.ac.uk

## Methods

### Cells

Abelson-transformed female and male mouse pre-B cells (mpre-B), derived from littermates, were generated in our lab from mouse foetal liver (E13.5) as described[73] and maintained in Iscove's modified Dulbecco's medium (IMDM) supplemented with 15% heat-inactivated fetal calf serum, 2 mM L-glutamine, 100 U ml$^{-1}$ penicillin, 100 µg ml$^{-1}$ streptomycin, 50 µM 2-mercaptoethanol and non-essential amino acids. MEFs used in this study were derived from E13.5 embryos of $Mof^{fl/fl}$ $Caag$ $ERT2$-cre mice[113] and Msl2 KO mice[109] as previously described[114] in collaboration with A. Akhtar's laboratory at the Max Planck Institute of Epigenetics and Immunobiology (Freiburg, Germany). Primary MEFs were cultured in standard Dulbecco's modified Eagle's medium (DMEM) supplemented with 10% heat-inactivated fetal calf serum, 2 mM L-glutamine, 100 U ml$^{-1}$ penicillin, 100 µg ml$^{-1}$ streptomycin, and 50 µM 2-mercaptoethanol, non-essential amino acids and 1 mM sodium pyruvate. All experiments were performed at passages 4–6.

All cell lines were maintained at 37 °C and 5% $CO_2$ in a humidified incubator. $Mof$ deletion was induced in $Mof^{fl/fl}$ $Cre$-$ERT2^{T/+}$ female MEFs by addition of 250 nM 4-OHT (4-hydroxytamoxifen; H6278, Sigma) for 3 days.

### Sex genotyping

Primers for genomic PCRs of $Sry$ and $Kdm5(c/d)$ (as described in refs. [115],[116]) were used to validate the sex of each cell line.

### WM-3835 treatment

For Kat7 (Hbo1) inhibition, mpre-B cells were seeded at a density of $5 \times 10^5$ cells per ml and treated with 1 µM WM-3835 (7366, Tocris Biosciences) or an equivalent volume of DMSO (vehicle) for 48 h.

### Metaphase spreads

Dividing cells were treated with 0.1 µg ml$^{-1}$ demecolcine (D1925, Sigma-Aldrich) to arrest cells in metaphase. Cells were pelleted, washed with PBS 1X and resuspended in freshly prepared hypotonic solution (75 mM KCl, pre-warmed to 37 °C) for 20 minutes at 37 °C. Nuclei were pelleted at 500g for 8 min and fixed in 3:1 MeOH:acetic acid (fixative solution), pre-cooled to −20 °C and stored overnight. The next day, samples were washed in fresh pre-cooled fixative solution, resuspended in a low volume of fixative and spread onto glass twin frost slides (onto a small drop of 45% acetic acid), air-dried and stored at room temperature (RT) until use. Chromosomes X or 3 were detected with XCyting Mouse Chromosome Painting Probes (Metasystems Probes), according to the protocol supplied by the manufacturer. Samples were mounted using Vectashield containing DAPI (Vector Laboratories). Leica SP5 II confocal microscope with LAS-AF software was used for imaging.

### Xist RNA-FISH

Xist cDNA (1 µg) was labelled with Cy3-dUTP (Cytiva) using a nick translation kit (Enzo LifeSciences) in a total volume of 50 µl. For every coverslip, 2.5 µl of labelled probe was ethanol precipitated with 1 µg salmon sperm DNA, and resuspended in 12 µl of 50% formamide, 2X saline sodium citrate (SSC), 10% dextran sulfate and 1 mg ml$^{-1}$ bovine serum albumin (BSA). Cells were fixed with 2.6% formaldehyde in PBS 1X for 10 min at room RT and permeabilized with 0.42% triton X-100 in PBS for 5 min on ice. Coverslips were then washed in PBS 1X, dehydrated with 70%–80%–95%–100% ethanol series and hybridized overnight in a humidified chamber at 37 °C. The following day coverslips were washed three times with 50% formamide/2X SSC at 42 °C and three times with 2X SSC at 42 °C. Coverslips were mounted in Vectashield with DAPI (Vector Laboratories) and imaged with an Olympus IX70 inverted microscope with a UPlanApo 100X/1.35 oil iris objective using Micro-Manager software (v.2.0).

### Immunofluorescence on cells

mpre-B cells were spun onto poly-L-lysine-coated slides by cytocentrifugation (Cytospin3, Shandon) at 163g for 10 min. Cells were fixed with 2% methanol-free formaldehyde (28906, Thermo Scientific) for 15 min and then permeabilized with 0.1% Triton X-100 for 10 min at RT. After blocking with 2% BSA/5% normal goat serum (blocking solution) for 1 h at RT, cells were incubated with H3K27me3-AF647 (1:100, 12158, Cell Signalling) or macroH2A1 (1:200, ab37264, Abcam) antibodies diluted in blocking solution at 4 °C overnight. For macroH2A1, cells were subsequently stained with goat anti-rabbit secondary antibody (Alexa Fluor 568, A-11011, Invitrogen) diluted 1:400 in blocking solution. Cells were nuclear stained with DAPI (1 µg ml$^{-1}$) for 5 min before mounting in Vectashield (Vector Laboratories). Images were taken with an Olympus IX70 inverted microscope with a UPlanApo 100X/1.35 oil iris objective using Micro-Manager software (v.2.0).

### Intracellular staining and flow cytometry analysis

For intracellular staining of acetylated histones, cells were fixed and permeabilised using the BD Cytofix/Cytoperm Fixation/Permeabilization Kit according to manufacturer's instructions. Cells were then incubated with conjugated antibodies diluted in permeabilization buffer for 1 h at RT. After washes, cells were resuspended in FACS buffer (PBS containing 2% FBS) and fluorescence acquired on BD FACSymphony A3 flow cytometer and BD FACSDiva Software (v.9.1), and data analysed using FlowJo software (v.10.8.1). The conjugated antibodies used were: H4K16ac-AF488 (1:100, 56999, Cell Signalling), H3K14ac-AF488 (1:200, ab277918, Abcam) and H4ac-AF488 (acetyl K5 + K8 + K12 + K16) (1:200, ab223995, Abcam).

### Mitotic chromosome preparation and flow sorting

Mitotic chromosome preparation and isolation by flow cytometry were performed as reported previously[40,66]. Cells were arrested in metaphase by adding 0.1 µg ml$^{-1}$ demecolcine solution (D1925, Sigma) to the culture medium. mpre-B cells were incubated with demecolcine solution for 6 h and MEFs for 9 h at 37 °C. Metaphase-arrested cells were collected and pelleted at 289g for 5 min and resuspended in 10 ml of hypotonic solution (75 mM KCl, 10 mM $MgSO_4$, 0.5 mM spermidine trihydrochloride and 0.2 mM spermine tetrahydrochloride, adjusted to pH 8) for 20 min at RT. Cells were then pelleted at 300 x g for 5 min at RT and resuspended in 1.5–3 ml of freshly prepared ice-cold polyamine isolation buffer (15 mM Tris-HCl, 2 mM EDTA, 0.5 mM EGTA, 80 mM KCl, 3 mM DTT, 0.25% Triton X-100, 0.2 mM spermine and 0.5 mM spermidine; pH 7.6-7.7). To release chromosomes, cells were vortexed at maximum speed for 30 s, syringed 3 to 8 times through a 21 G needle and centrifuged at 200 x g for 2 min at 4 °C. The supernatant containing chromosomes was collected and passed through a 20 µm mesh Cell-Trics filter (04-0042-2315, Sysmex). Chromosomes were stained at 4 °C overnight with H3K27me3 AF-647 antibody (12158, Cell Signalling, 1:50) and the next day with 5 µg/ml Hoechst 33258, 25 µg/ml chromomycin A3 and 10 mM $MgSO4$ for 1 h at RT. Sodium citrate (10 mM final) and sodium sulfite (25 mM final) were added to chromosome suspensions 1 h prior to analysis and flow-sorting.

Individual chromosome populations were purified using BD Influx with BD FACS software (v.1.2.0.142). Hoechst 33258 was excited using a 355 nm laser (350 mW) and fluorescence was collected using a 400 nm long pass filter in combination with a 500 nm short pass filter. Chromomycin A3 was excited using a 457 nm laser (300 mW) and fluorescence was collected using a 500 nm long pass filter in combination with a 600 nm short pass filter. For isolating Xi and Xa, H3K27me3-AF647 was excited using a 637 nm laser (160 mW), and the resulting fluorescence was collected using a 660/30 bandpass filter. Forward scatter was measured using a 488 nm laser (200 mW). Chromosomes were sorted using a 70 µm nozzle tip, with a drop drive frequency of ~96 kHz and sheath pressure at 65 psi and were collected into FACS tubes containing ice-cold polyamine buffer.

## Chromosome painting on flow-sorted chromosomes

Flow-sorted chromosomes X ($10^5$) were spun onto poly-L-lysine-coated slides (VWR) by cytocentrifugation (Cytospin3, Shandon) at 163$g$ for 10 min at RT, followed by fixation in 3:1 MeOH:acetic acid (fixative) for 30 min at −20 °C. Samples were air-dried and hybridised with mouse chromosome X or 19 paint (Metasystem probes) according to the manufacturer's instructions.

## Immunofluorescence on flow-sorted chromosomes

Flow-sorted chromosomes X, 19 or 3 were cytospun onto poly-L-lysine coated slides (Cytospin3, Shandon) at 163$g$ for 10 min, RT. Cytospun mitotic chromosomes (unfixed) were incubated with blocking buffer (3% normal goat serum in 10 mM Hepes, 2 mM $MgCl_2$, 100 mM KCl and 5 mM EGTA) for 30 min at RT. Chromosomes were stained (overnight at 4 °C or 2 h at RT) in a humid chamber with primary antibodies to macroH2A1 (1:100, ab37264, Abcam), Trim28 (1:100, ab10484, Abcam), Kat7 (1:50, ab70183, Abcam), Noc2L (1:50, PA5-101730, Invitrogen), H3K14ac (1:200, ab52946, Abcam), H4K16ac (1:200, ab109463, Abcam), H4ac (1:200, 06-598, Merck Millipore). Samples were then washed and incubated with goat anti-rabbit secondary antibody (Alexa 488, A11034, Invitrogen) diluted 1:400 in blocking buffer for 1 h at RT. Samples were mounted in Vectashield mounting medium containing DAPI (Vector Laboratories). Images were taken with an Olympus IX70 inverted microscope with a 100X oil objective using Micro-Manager software.

## Size measurements of flow-sorted chromosomes

Flow-sorted chromosomes were spun onto poly-L-lysine-coated slides by cytocentrifugation at 163$g$ for 10 min and mounted in Vectashield containing DAPI (Vector Laboratories). Chromosome images were acquired on an Olympus IX70 inverted microscope with a UPlanApo 100X/1.35 oil iris objective using Micro-Manager software (v.2.0). Chromosome size measurements were performed by estimating whole chromosome areas on ImageJ.

## Proteomics and data processing

Proteomics for sorted chromosomes X and 19 or Xi and Xa and subsequent data analyses were performed using a method previously described[40,66] with some modifications. Proteomic analysis of sorted chromosome X and 19 from female and male cells were carried out in six biological replicates and analysis of Xi, Xa and X chromosomes isolated from male cells were performed in a minimum of 3 biological replicates. Sorted chromosomes were pelleted by centrifugation at 13,000$g$ for 15 min at 4 °C and snap frozen at −80 °C. Samples were digested with trypsin using the iST Sample Preparation Kit (PreOmics, P.O.00001), according to the manufacturer's recommendations.

Resuspended peptide digests were analysed by LC-MS/MS analysis on an Ultimate 3000 RSLC nano liquid chromatography system (Thermo Scientific) coupled to a Q-Exactive HFX mass spectrometer (Thermo Scientific) via an EASY spray source (Thermo Scientific). Samples were loaded onto a trap column (Acclaim PepMap 100 C18, 100 μm × 2 cm) for desalting and concentration at a flow rate of 8 μl min$^{-1}$ (loading pump buffer: 2% acetonitrile, 0.1% TFA). Peptides were then eluted on-line to an analytical column (Acclaim Pepmap RSLC C18, 75 μm × 50 cm) at a flow rate of 250 nl min$^{-1}$. Peptides were separated using a 120 min gradient, 1–25 % of buffer B for 90 min followed by 25–45% buffer B for another 30 min (buffer A: 5% DMSO, 0.1% FA; buffer B: 75% acetonitrile (ACN), 20% water, 5% DMSO, 0.1 % FA) and subsequent column conditioning and equilibration. Eluted peptides were analysed by the mass spectrometer operating in positive polarity using a data-dependent acquisition (DDA) mode. Ions for fragmentation were determined from an initial MS1 survey scan at 120,000 resolution followed by HCD (Higher Energy Collision Induced Dissociation) fragmentation of the top 25 most abundant ions at 15,000 MS2 resolution. MS1 and MS2 scan AGC targets were set to 3e6 and 5e4, allowing maximum ion injection times of 25 ms and 85 ms, with

spectrum data types set to profile and centroid, respectively. A survey scan $m/z$ range of 350–1,750 was used, normalized collision energy set to 27%, charge state exclusion enabled with unassigned and +1 charge states rejected and a minimal AGC target of 8e3. Dynamic exclusion was set to 30 seconds.

Data were processed using the MaxQuant software platform (v.1.6.10.43)[117], with database searches carried out by the in-built Andromeda search engine against the relevant Swissprot protein databases for *Mus Musculus* (downloaded – 4 January 2018, entries: 16,958) reverse decoy database approach was used at a 1% false-discovery rate (FDR) for peptide spectrum matches and protein identifications. Search parameters included: maximum missed cleavages set to 2, variable modifications of oxidation (M), acetyl (protein N-term), acetyl (K), trimethyl (K), and GlyGly (K). Fixed modification of cysteine carbamidomethylation was used. Label-free quantification was enabled with an LFQ minimum ratio count of 1. The 'match between runs' function was enabled, with alignment and matching time windows of 20 and 0.7 minutes respectively.

Statistical analysis and data visualization were performed using the Perseus software platform (v.1.6.7.0 or v.1.6.15.0)[118]. The protein-Groups.txt file was analysed by uploading the data matrix with the respective LFQ intensities as main columns. The data matrix was filtered based on categorical columns to remove reverse decoy hits, potential contaminants and protein groups which were 'only identified by site'. Missing values were replaced with 'NaN', and data were log transformed. Biological replicates were grouped and conditions were defined in categorical annotation rows. Volcano plots were generated based on LFQ intensities with the following settings: test: $t$-test; side: both; number of randomisations: 250; preserve grouping in randomisations: <none>; FDR: 0.05; S0: 0.1.

Sites tables for the variable modification acetyl (K) was analysed on the Perseus platform (v.1.6.15.0). Sites tables underwent the site expansion process in Perseus. Briefly, sample columns that contain the suffixes "_1, _2, _3", alongside identifier columns, were loaded into the Perseus environment. These columns represent the intensity of a modified peptide precursor that is either singly, doubly or triply modified. A single sample is then defined as three columns, with the structure repeated for every sample. Sites expansion is carried out in the Perseus interface, in the "Modifications/Expand site table" section of the tool bar. The resulting table structure returns a single column per sample, with the modification status of the precursor defined by a new column called "multiplicity". The multiplicity column has the entries "_1, _2, _3" as rows, for every modified peptide precursor. Therefore, a singly ("_1") modified peptide will be 1 row with intensity reported in the sample name columns. A doubly ("_2") modified peptide will have 2 rows, with the same intensity reported in both rows. The same convention is followed for triply ("_3") modified peptides. Following sites expansion, the data were filtered to remove potential contaminants and to keep sites with minimum one intensity per site.

## DNA methylation analysis (bisulfite-PCR)

DNA was purified from sorted Xi or Xa chromosomes using the DNA miniprep plus kit (Zymo) and bisulfite-converted using the EZ DNA methylation kit (Zymo) in accordance with manufacturer's protocols. After PCR amplification of CGIs of interest (nested in the case of *Xist* CGI), products were cloned using the CloneJET PCR Cloning Kit (ThermoFisher Scientific) and sequenced. Results were analysed using BiQ Analyser[119] and plotted using Methylation plotter[120]. Primers used for amplification of CGIs of *Hprt*, *Pdk3*, *Eif2s3x* and *Xist* have been described in refs. [121,122].

## Immunostaining and flow cytometry analysis of flow-sorted chromosomes

Flow-sorted chromosomes were stained with H3K27me3-AF647 antibody (1:50, 12158, Cell Signalling) in combination with the indicated

acetylated histone antibodies for 1 h at RT and re-analysed on BD Influx with BD FACS software (v.1.2.0.142). The acetylated histone antibodies used are as follows: H4K16ac-AF488 (1:50, 56999, Cell Signalling), H3K14ac-AF488 (1:100, ab277918, Abcam) and H4ac-AF488 (1:100, ab223995, Abcam). Data acquired were analysed and histograms were plotted using FlowJo software (v.10.8.1).

### Real-time quantitative PCR
RNA was extracted from cells using the RNeasy Mini kit (Qiagen), followed by the depletion of genomic DNA with the TURBO DNA-free kit (Invitrogen). Reverse transcription was carried out with random primers and Superscript III Reverse Transcriptase (Invitrogen). Quantitative real-time PCR was conducted on a CFX96 Real-Time System (Bio-Rad, CFX Manager v.3.1) using QuantiTect SYBR Green Master Mix (Qiagen) in a 10 μl reaction volume, with primers specified in Supplementary Table 8.

### Statistics and reproducibility
Statistical analysis and visualisation of proteomic data were performed in Perseus (v.1.6.7.0 or v.1.6.15.0) using unpaired two-tailed Student's $t$-tests with permutation-based FDR correction (FDR < 0.05). All other statistical analyses were performed using unpaired two-tailed $t$-test and graphs were plotted in Microsoft Excel or GraphPad Prism. A $P$ value < 0.05 was considered statistically significant. Experiments were independently repeated at least three times. Where representative images are shown, data were highly consistent across three independent experiments. Proteomic analyses were conducted using a minimum of three biological replicates. Chromosome size measurements were performed on a minimum of 85 chromosomes for mpre-B cells and at least 25 chromosomes for primary fibroblasts, with data collected over three independent experiments.

### Reporting summary
Further information on research design is available in the Nature Portfolio Reporting Summary linked to this article.

### Data availability
The mass spectrometry proteomics data have been deposited to the ProteomeXchange Consortium via the PRIDE partner repository with the following dataset identifier PXD054014, and are publicly available. Gene Ontology annotations for use in Perseus were downloaded from http://annotations.perseus-framework.org (mainAnnot.mus_musculus.txt). All other relevant data supporting the key findings of this study are available within the article and Supplementary data files provided or from the corresponding author upon request. Source data are provided with this paper.

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

### Acknowledgements
We are grateful to the LMS and Imperial NIHR Flow Cytometry Facility, LMS proteomic facility and LMS microscopy facility for their support. This work was funded by the Medical Research Council UK to A.G.F. (MC PC 23014 and MC 22015) and A.G.F. and M.M. (MC 20027516, MC_UP_1605/12 and MC_UP_1605/11). D.D. received an International Strategic Support Fund from the Wellcome Trust (WCMA_PSN102).

### Author contributions
A.G.F. and D.D. designed the study. D.D. and S.C. performed most of the experiments, conducted data analysis and designed the figures. A.G.F., D.D. and S.C. wrote the manuscript. B.P. conducted mitotic chromosome flow sorting. H.K. and A.M. conducted MS and helped with proteomic analysis. K.E.B. conducted FISH and metaphase spread experiments. C.W. helped with microscopy and image analysis. T.B.N. and G.W. helped with additional experiments and genome-wide analysis. I.G. and R.K. derived Msl2 and Mof inducible knockout cells. M.M., N.B. and A.A. provided scientific advice and support with experiments.

### Competing interests
The authors declare no competing interests.

### Additional information
**Extended data** is available for this paper at https://doi.org/10.1038/s41556-025-01748-0.

**Correspondence and requests for materials** should be addressed to Dounia Djeghloul or Amanda G. Fisher.

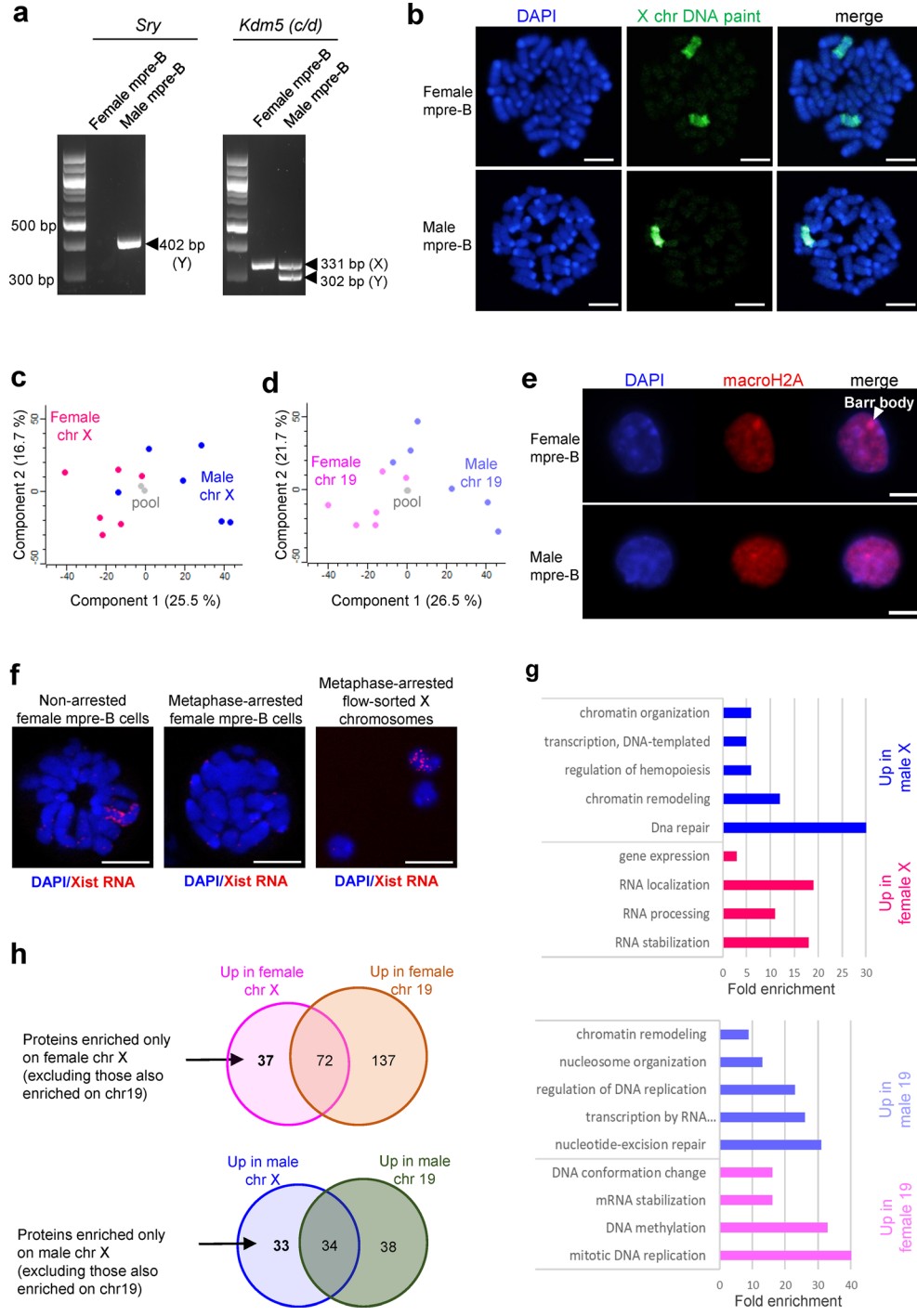

**Extended Data Fig. 1 | Comparison of mitotic chromosome proteome from female and male mpreB cells.** Extended data Fig. 1 (related to Fig. 1).
**(a)** Detection of *Sry* and *Kdm5 (c/d)* genes in genomic DNA isolated from female and male mpre-B cells, by PCR. A PCR product (402 bp) corresponding to *Sry* gene was detected uniquely in male mpre-B cells. *Kdm5* gene has X- and Y-specific variants (*Kdm5c/d*) resulting in two bands in male cells (302 and 331 bp) and one band in female cells (331 bp). **(b)** DNA FISH analysis showing X chromosome-specific paint (green) labelling metaphase chromosomes in XY male and XX female mpre-B cells. DAPI is shown in blue. Scale bars = 5 μm. **(c, d)** Principal component analysis (PCA) of proteomic data derived from female and male mitotic X chromosomes (**c**) and female and male mitotic chromosome 19 (**d**). Each dot corresponds to an individual replicate (n = six biological replicates).

**(e)** Representative images of interphase female and male mpre-B cells stained with antibody against macroH2A1 (red). DAPI is shown in blue. Scale bars = 4 μm. **(f)** Xist RNA FISH of 'non-arrested' metaphase female mpre-B cells, metaphase-arrested (demecolcine-treated) cells and flow sorted X chromosomes derived from metaphase-arrested cells. Scale bars = 5 μm. **(g)** GO term analysis of proteins significantly enriched in male versus female mitotic X chromosome samples (top panel) or male versus female mitotic chromosome 19 samples (lower panel). **(h)** Venn diagram (upper) showing overlap in factors significantly enriched on female mitotic X chromosomes and female mitotic chromosome 19, as compared to their male equivalents. Lower diagram shows overlap in factors significantly enriched on male mitotic X chromosomes and male mitotic chromosome 19, as compared to their female equivalents.

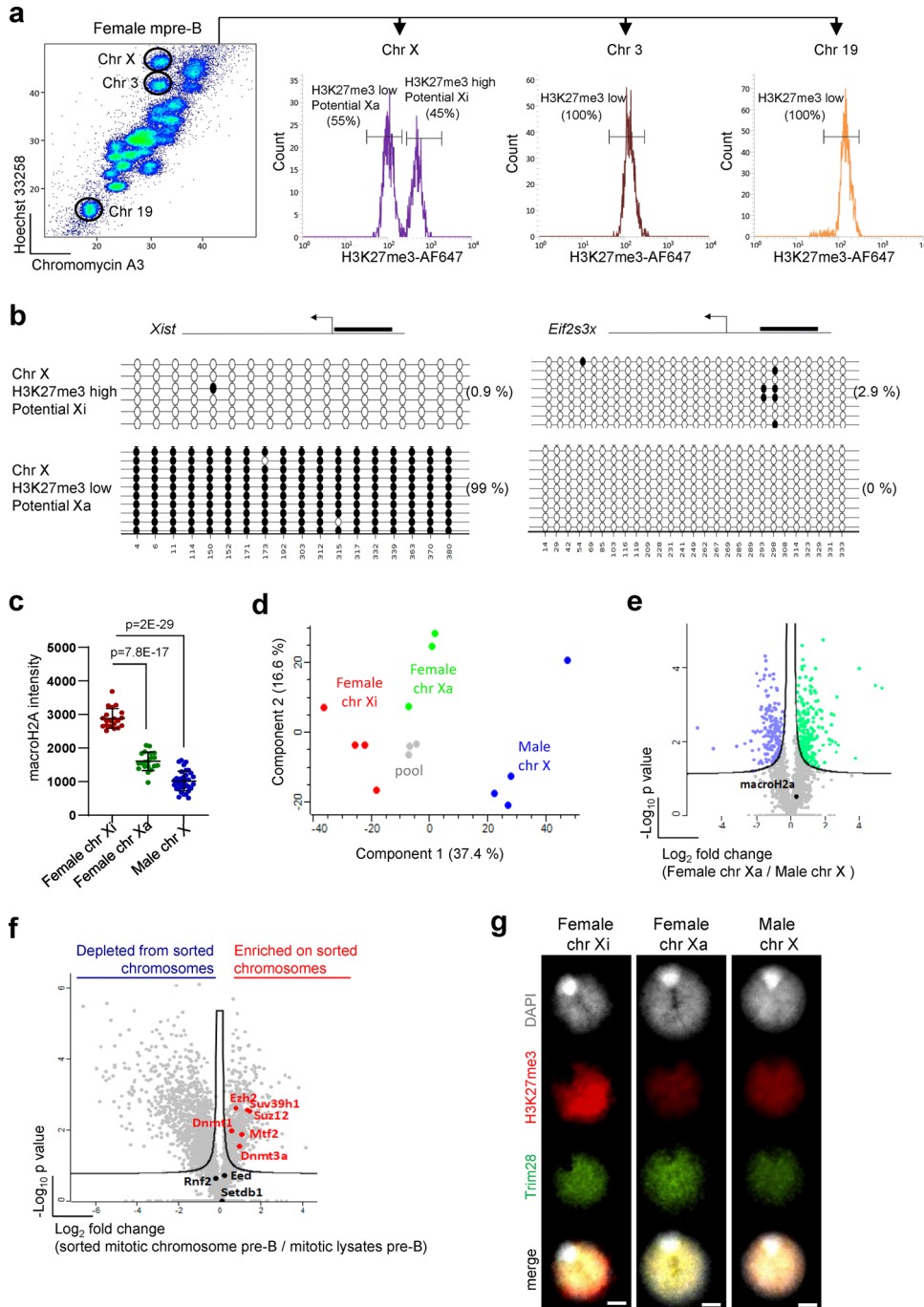

**Extended Data Fig. 2 | Flow-sorting validation and proteomic analysis of mitotic Xi and Xa chromosomes.** Extended data Fig. 2 (related to Fig. 2).
**(a)** H3K27me3 staining profile of mitotic X, 3, and 19 chromosomes isolated from female mpre-B cells gated according to the bivariate karyotype. **(b)** Bisulfite analysis of CpG DNA methylation in H3K27me3 high and H3K27me3 low flow-purified X chromosomes to validate their Xi and Xa epigenetic status. Diagrams show the location of the region analysed relative to the gene transcription start site (arrows) for *Xist* (left panels) and *Eif2s3x* (right panels) genes. Each circle represents a CpG dinucleotide and each line represents methylation on an individual DNA strand determined by sequencing subcloned PCR product from bisulfite-treated genomic DNA. Black circles = methylated CpGs, White circles = unmethylated CpGs. Percentage of methylation is indicated in brackets. **(c)** Quantification of macroH2A mean fluorescence intensity of sorted mitotic Xa, Xi, and male X chromosomes. Number of chromosomes analysed: 20, 20 and 36 for Xi, Xa, and male X chromosome respectively. P-values of statistically significant changes relative to Xi, measured by unpaired two-tailed Student's t-tests, are indicated. p = 7.8E-17 and p = 2E-29. **(d)** Principal component analysis (PCA) of proteomic data derived from female mitotic Xi, Xa, male chromosome

X and pooled samples. Each dot represents an individual replicate. **(e)** Volcano plot showing proteins differentially enriched between female chromosome Xa and male chromosome X. MacroH2A is not significantly changed between female Xa and male X. Volcano plot was generated on Perseus software using unpaired two-tailed Student's t-test, permutation-based FDR < 0.05, s0 = 0.1. **(f)** Volcano plot showing proteins detected as being significantly enriched or depleted on sorted mitotic chromosomes relative to mitotic lysate pellet of female mpre-B cells (unpaired two-tailed Student's *t*-test, permutation-based FDR < 0.05), n = three biological replicates each measured in duplicate. Proteins were plotted as Log2 fold change (LFQ intensity of sorted chromosome pellet /LFQ intensity of mitotic lysate pellet) and significance (−Log10 p-value) using Perseus software. Chromatin repressors significantly enriched on mitotic chromosomes are highlighted in red, not significantly enriched shown in black and those depleted from mitotic chromosomes in blue. **(g)** Immunofluorescence images of Trim28 (green) on flow-sorted mitotic female Xi, female Xa, and male X chromosomes isolated from female and male mpre-B cells. DAPI counterstain is shown in light grey. Scale bars = 2 μm. Source data is available for Extended Data Fig. 2c.

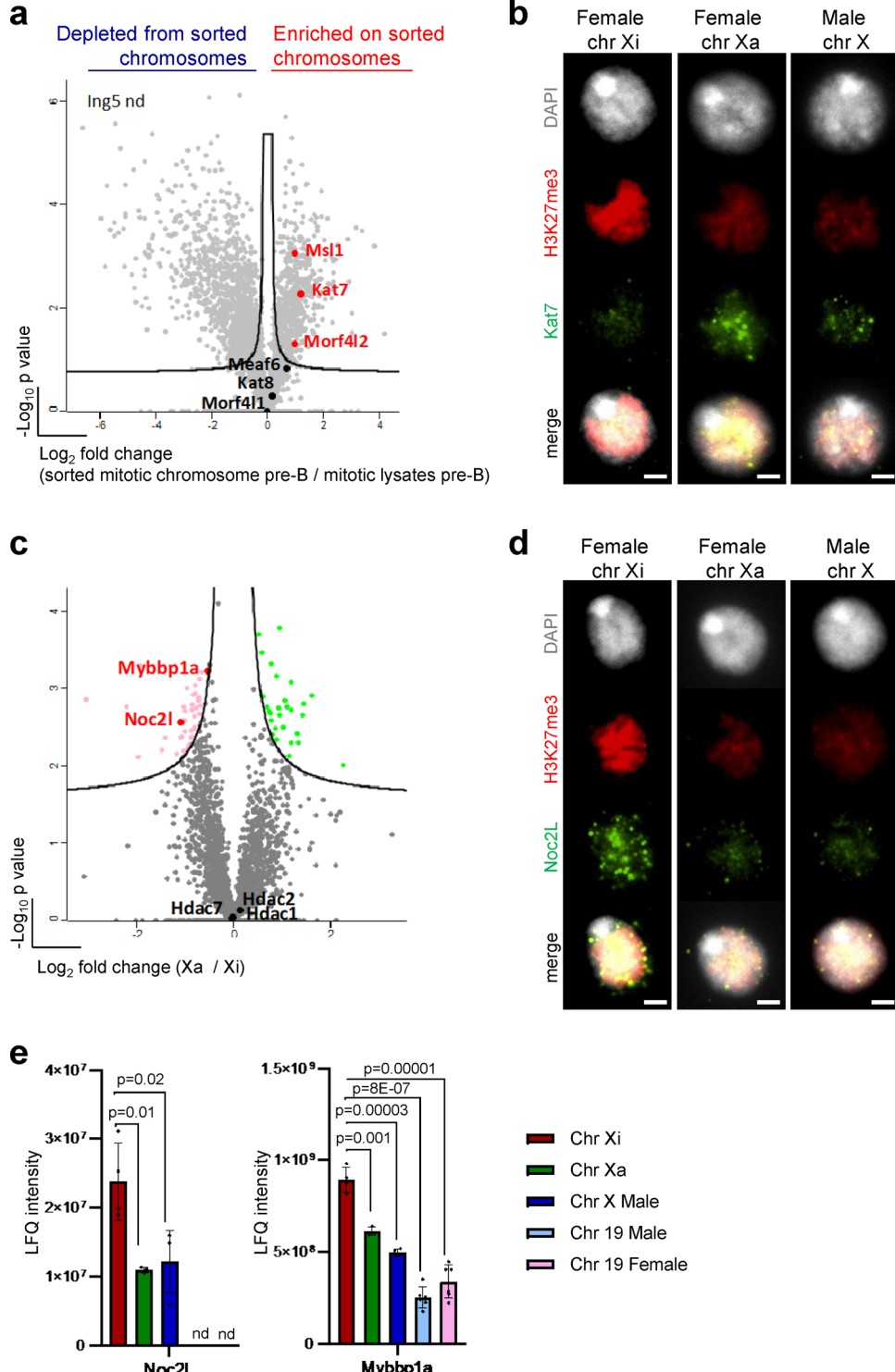

**Extended Data Fig. 3 | See next page for caption.**

**Extended Data Fig. 3 | Differences in Kat7, Noc2l and Mybbp1a abundance on mitotic X chromosomes.** Extended data Fig. 3 (related to Fig. 3). **(a)** Volcano plot as in extended data Fig. 2e. HAT components significantly enriched on mitotic chromosomes are highlighted in red, and those not significantly enriched are shown in black. **(b)** Immunofluorescence images of Kat7 (green) on flow-sorted mitotic female Xi (H3K27me3 high, in red), female Xa (H3K27me3 low, in red), and male X (H3K27me3 low, in red) chromosomes isolated from female and male mpre-B cells. DAPI counterstain is shown in light grey. Scale bars = 2 μm. **(c)** Volcano plot of proteins significantly enriched on female Xi (red) and Xa (green) isolated from mpre-B cell lines. Unpaired two-tailed Student's t-test, permutation-based FDR < 0.05, s0 = 0.1, n = four biological replicates of Xi and three for Xa. Proteins were plotted as Log2 fold change (LFQ intensity of flow-purified mitotic Xa over Xi chromosome) versus significance (-Log10 of p-value) using Perseus software. Inhibitors of HAT activity or HDACs enriched on Xi (red) or not significantly different (black) between mitotic Xi and Xa are highlighted. **(d)** Immunofluorescence images of Noc2l (green) on flow-sorted mitotic female Xi (H3K27me3 high, in red), female Xa (H3K27me3 low, in red), and male X (H3K27me3 low, in red) chromosomes isolated from female and male mpre-B cells. DAPI counterstain is shown in light grey. Scale bars = 2 μm. **(e)** Average LFQ intensity of Noc2L and Mybbp1a in female chromosome Xi, Xa, male chromosome X, male and female chromosome 19 samples. Mean ± SD is shown, n = 4 biological replicates for Xi and male X, three for female Xa, and six for chromosome 19. P-value of statistically significant changes relative to Xi, measured by unpaired two-tailed Student's t-tests, is indicated. nd = not detected. p = 0.01 and p = 0.02 (Noc2L, left plot). p = 0.001, p = 0.00003, p = 8E-07, and p = 0.00001 (Mybbp1a, right plot). Source data is available for Extended Data Fig. 3e.

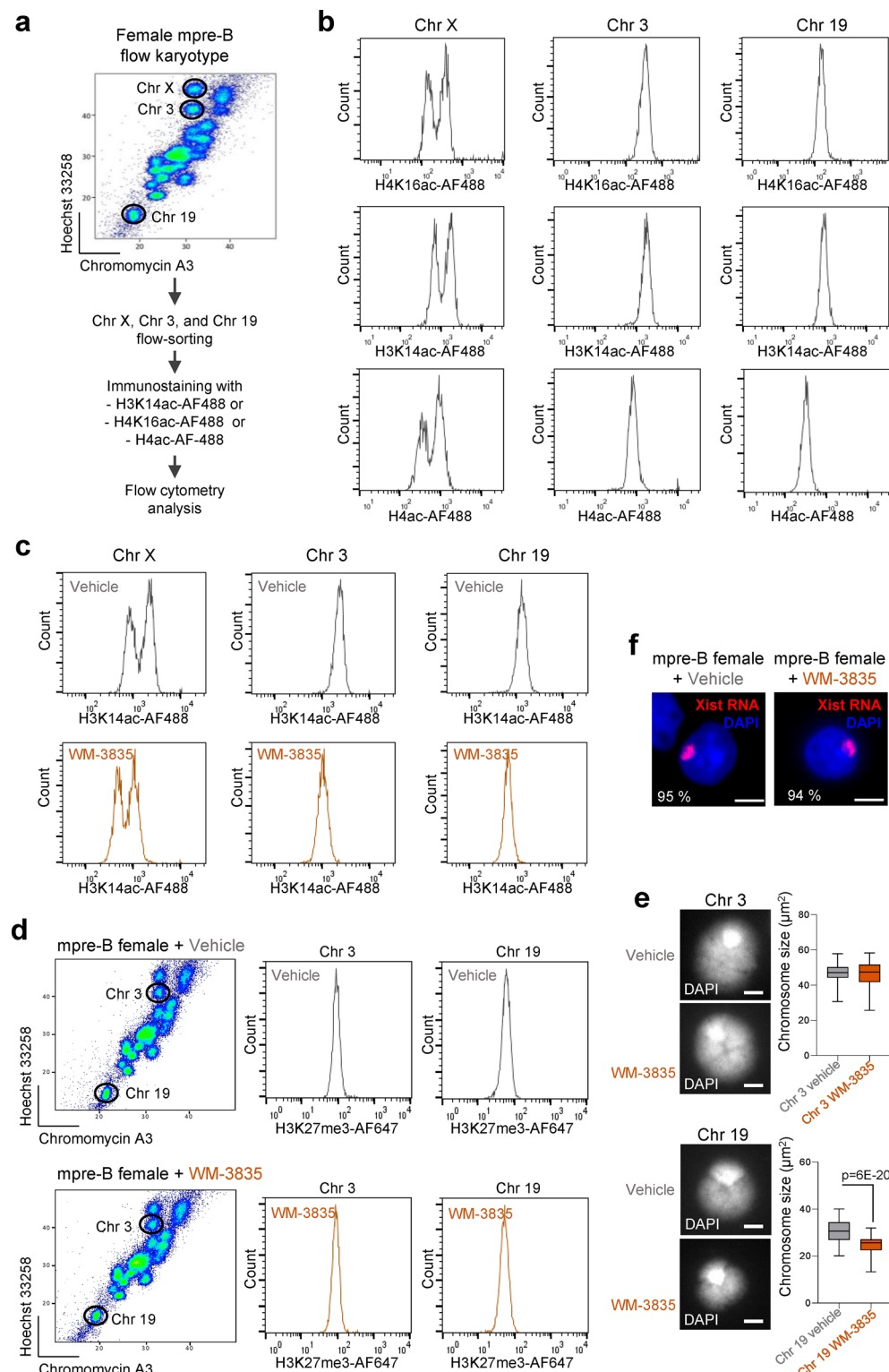

**Extended Data Fig. 4 | Effects of Kat7 inhibition on mitotic chromosomes in female mpre-B cells.** Extended data Fig. 4 (related to Fig. 4). **(a)** Experimental procedure used for analysis of histone acetylation modifications on mitotic chromosomes X, 3 and 19 isolated from female mpre-B cells. **(b)** H3K14ac, H4K16ac and pan-H4ac immunostaining and flow cytometric analysis of female-derived mitotic chromosomes X, 3 and 19. **(c)** H3K14ac immunostaining and flow cytometric analysis of mitotic chromosomes X, 3, and 19 isolated from female mpre-B cells treated with DMSO vehicle or WM-3835. **(d)** H3K27me3 labelling intensity of chromosomes 3 and 19 isolated from female mpre-B cells treated with DMSO or WM-3835. Histograms shown are representative of 3 independent experiments. **(e)** Representative images of flow-sorted mitotic chromosomes 3 and 19 sorted from female mpre-B cells treated with DMSO or

WM-3835. DAPI stain in light grey. Scale bars = 2 μm. Plot (right of the images) show corresponding chromosome size measurements for mitotic chromosomes 3 and 19 in each condition. Minimum, lower quartile, median, upper quartile and maximum values are indicated. Number of chromosomes analysed: 102, 97 for chromosome 3 and 108, 100 for chromosome 19 vehicle or WM-3835-treated respectively. Data collected over three independent experiments. P-values of statistically significant changes, measured by unpaired two-tailed Student's t-tests, are indicated. p = 6E-20 (right lower plot). **(f)** Xist RNA FISH (red) images of interphase female mpre-B cells treated with DMSO or WM-3835, DAPI stain is shown in blue. Scale bars = 4 μm. Percentage of cells with one Xist RNA domain is indicated. Source data is available for Extended Data Fig. 4e.

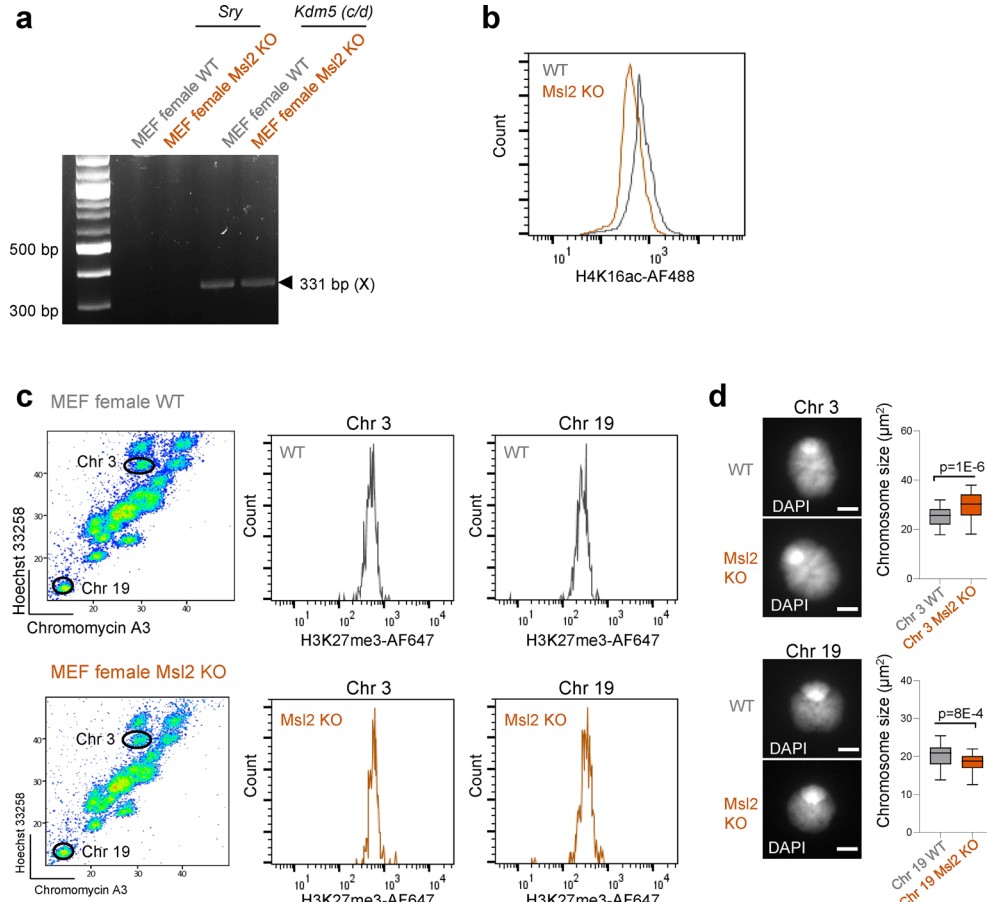

**Extended Data Fig. 5 | H4K16ac levels and mitotic chromosome sizes in female Msl2 KO MEFs.** Extended data Fig. 5 (related to Fig. 5 a-c). **(a)** Genomic PCR analysis of *Sry* and *Kdm5 (c/d)* genes in WT and Msl2 KO female MEFs. **(b)** H4K16ac staining and flow cytometric analysis of WT and Msl2 KO female MEFs. Histograms shown are representative of three independent experiments. **(c)** Hoechst 33258 and Chromomycin A3 bivariate mitotic chromosome flow karyotype (left panels) of WT and Msl2 KO female MEFs is shown. Gates used to sort chromosome 3 and 19 are indicated. H3K27me3 staining profiles of mitotic chromosomes 3 (middle panel) and chromosome 19 (right panel) isolated from WT (top panel) and Msl2 KO (lower panel) female MEFs. **(d)** Representative images (left) and size measurements (right) of mitotic chromosomes 3 and 19 flow-sorted from WT and Msl2 KO female MEFs. DAPI counterstain is shown in light grey. Scale bars = 2 μm. For chromosome size measurements: Minimum, lower quartile, median, upper quartile and maximum values are indicated. Number of chromosomes analysed: 62, 41 for chromosome 3 and 45, 47 for chromosome 19 WT or Msl2 KO respectively. Data collected over three independent experiments. P-values of statistically significant changes, measured by unpaired two-tailed Student's t-tests, are indicated. p = 1E-6 (Chr3, right upper plot) and p = 8E-4 (Chr19, right lower plot). Source data is available for Extended Data Fig. 5d.

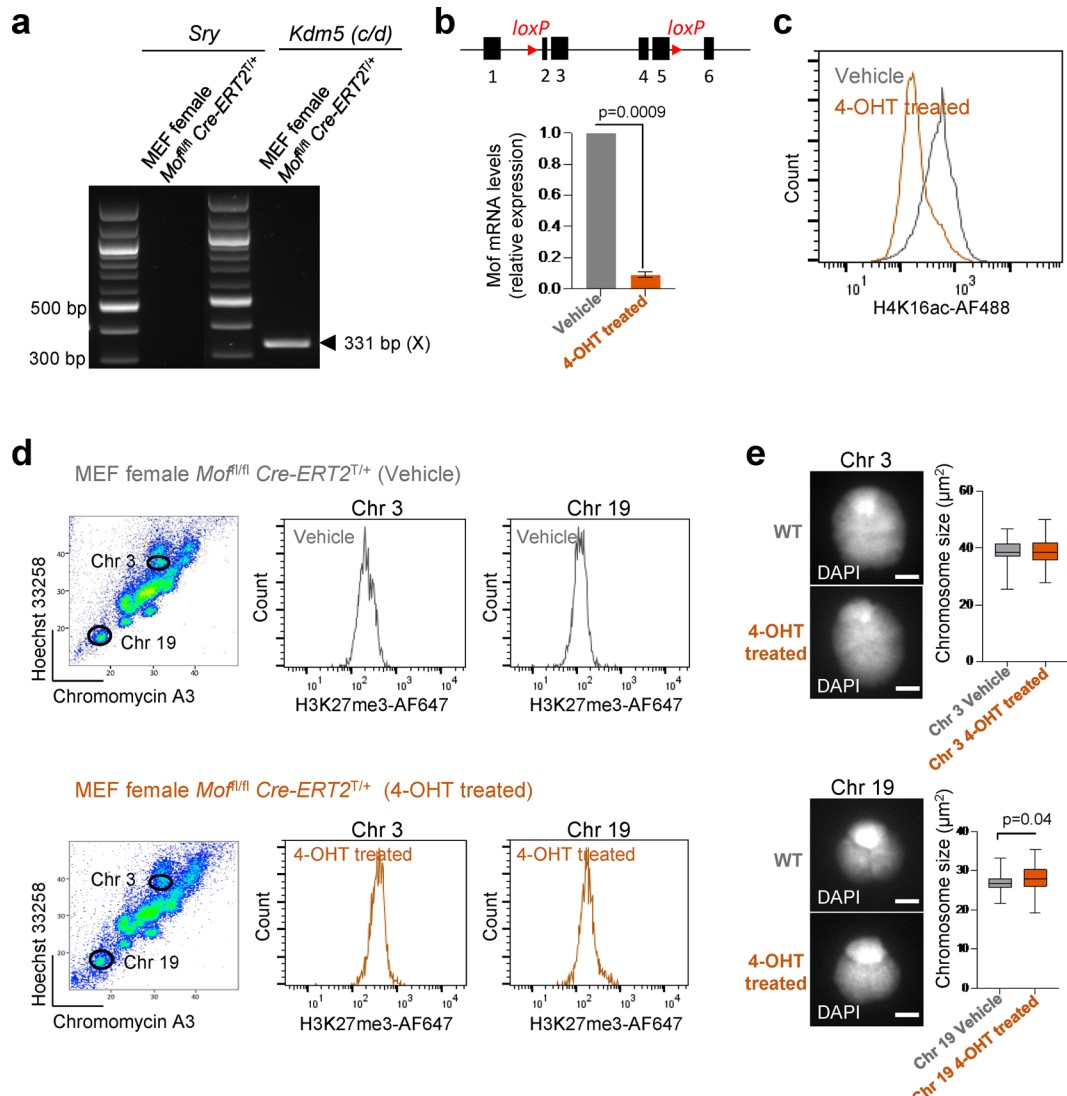

**Extended Data Fig. 6 | H4K16ac levels and mitotic chromosome sizes in female Mof conditional KO MEFs.** Extended data Fig. 6 (related to Fig. 5 d-i). **(a)** Genomic PCR analysis of *Sry* and *Kdm5 (c/d)* genes in *Mof*^fl/fl *Cre-ERT2*^T/+ female MEFs. **(b)** *Mof* transcript levels in *Mof*^fl/fl *Cre-ERT2*^T/+ female MEFs treated with 4-OHT or vehicle for 3 days measured by RT-qPCR after normalisation to *β-Actin* transcript levels (n = three independent experiments; mean ± SD shown). P-value of statistically significant decrease, measured by unpaired two-tailed Student's t-test on delta-Ct values is indicated. p = 0.0009. Diagram (top) shows location of the loxP sites in the *Mof* gene. **(c)** H4K16ac staining and flow cytometric analysis of *Mof*^fl/fl *Cre-ERT2*^T/+ female MEFs treated with 4-OHT or vehicle for 3 days. Histograms shown are representative of 4 independent experiments. **(d)** Hoechst 33258 and Chromomycin A3 bivariate mitotic chromosome flow karyotype (left panels) of *Mof*^fl/fl *Cre-ERT2*^T/+ female MEFs treated with 4-OHT or vehicle

for 3 days. Gates used to sort chromosomes 3 and 19 are indicated. H3K27me3 staining profile (right panels) of mitotic chromosomes 3 and 19 isolated from *Mof*^fl/fl *Cre-ERT2*^T/+ female MEFs treated with 4-OHT or vehicle for 3 days. **(e)** Representative images (left) and size measurements (right) of mitotic chromosomes 3 and 19 flow-sorted from *Mof*^fl/fl *Cre-ERT2*^T/+ female MEFs treated with 4-OHT or vehicle. DAPI counterstain is shown in light grey. Scale bars = 2 μm. For chromosome size measurements: Minimum, lower quartile, median, upper quartile and maximum values are indicated. Number of chromosomes analysed: 57, 47 for chromosome 3 and 48, 42 for chromosome 19 vehicle or 4-OHT-treated respectively. Data collected over three independent experiments. P-values of statistically significant decreases, measured by unpaired two-tailed Student's t-tests, are indicated. p = 0.04 (Chr19, right lower plot). Source data is available for Extended Data Fig. 6b and e.

| | |
|---|---|

# Reporting Summary

## Statistics

For all statistical analyses, confirm that the following items are present in the figure legend, table legend, main text, or Methods section.

| n/a | Confirmed | |
|---|---|---|
| ☐ | ☒ | The exact sample size (*n*) for each experimental group/condition, given as a discrete number and unit of measurement |
| ☐ | ☒ | A statement on whether measurements were taken from distinct samples or whether the same sample was measured repeatedly |
| ☐ | ☒ | The statistical test(s) used AND whether they are one- or two-sided<br>*Only common tests should be described solely by name; describe more complex techniques in the Methods section.* |
| ☐ | ☒ | A description of all covariates tested |
| ☒ | ☐ | A description of any assumptions or corrections, such as tests of normality and adjustment for multiple comparisons |
| ☐ | ☒ | A full description of the statistical parameters including central tendency (e.g. means) or other basic estimates (e.g. regression coefficient) AND variation (e.g. standard deviation) or associated estimates of uncertainty (e.g. confidence intervals) |
| ☐ | ☒ | For null hypothesis testing, the test statistic (e.g. *F*, *t*, *r*) with confidence intervals, effect sizes, degrees of freedom and *P* value noted<br>*Give P values as exact values whenever suitable.* |
| ☒ | ☐ | For Bayesian analysis, information on the choice of priors and Markov chain Monte Carlo settings |
| ☒ | ☐ | For hierarchical and complex designs, identification of the appropriate level for tests and full reporting of outcomes |
| ☒ | ☐ | Estimates of effect sizes (e.g. Cohen's *d*, Pearson's *r*), indicating how they were calculated |

*Our web collection on statistics for biologists contains articles on many of the points above.*

## Software and code

Policy information about availability of computer code

| Data collection | BD FACS software (vl.2.0.142) and BD DIVA (v9.1) were used to collect flow cytometry data. Micro-Manager (v2.0, Olympus IX70 microscope) and LAS-AF (2.7.3.9723, SPS II microscope) were used to collect imaging data. Quantitative real-time PCR data was collected using Bio-Rad CFX Manager software (v3.1). |
|---|---|
| Data analysis | Flow cytometry data was analysed using FlowJo software (v10.8.1). ImageJ/Fiji (version 1.54e) was used for image analysis including mitotic chromosome size measurements. Proteomic data was analysed using the Label-Free Quantification algorithm in the MaxQuant software platform (v1.6.10.43). The Perseus software (vl.6.7.0 and v1.6.15.0) was used for both statistical analysis and data visualisation of the proteomics data. Gene Ontology analysis was performed at http://geneontology.org/. |

For manuscripts utilizing custom algorithms or software that are central to the research but not yet described in published literature, software must be made available to editors and reviewers. We strongly encourage code deposition in a community repository (e.g. GitHub). See the Nature Portfolio guidelines for submitting code & software for further information.

## Data

Policy information about availability of data

All manuscripts must include a data availability statement. This statement should provide the following information, where applicable:

- Accession codes, unique identifiers, or web links for publicly available datasets
- A description of any restrictions on data availability
- For clinical datasets or third party data, please ensure that the statement adheres to our policy

The mass spectrometry proteomics data have been deposited to the ProteomeXchange Consortium via the PRIDE partner repository with the dataset identifier PXD054014. Gene Ontology annotations for use in Perseus were downloaded from http://annotations.perseus-framework.org (mainAnnot.mus_musculus.txt). All other relevant data supporting the key findings of this study are available within the article and Supplementary Tables provided in the submission.

## Research involving human participants, their data, or biological material

Policy information about studies with human participants or human data. See also policy information about sex, gender (identity/presentation), and sexual orientation and race, ethnicity and racism.

| | |
|---|---|
| Reporting on sex and gender | *Use the terms sex (biological attribute) and gender (shaped by social and cultural circumstances) carefully in order to avoid confusing both terms. Indicate if findings apply to only one sex or gender; describe whether sex and gender were considered in study design; whether sex and/or gender was determined based on self-reporting or assigned and methods used. Provide in the source data disaggregated sex and gender data, where this information has been collected, and if consent has been obtained for sharing of individual-level data; provide overall numbers in this Reporting Summary. Please state if this information has not been collected. Report sex- and gender-based analyses where performed, justify reasons for lack of sex- and gender-based analysis.* |
| Reporting on race, ethnicity, or other socially relevant groupings | *Please specify the socially constructed or socially relevant categorization variable(s) used in your manuscript and explain why they were used. Please note that such variables should not be used as proxies for other socially constructed/relevant variables (for example, race or ethnicity should not be used as a proxy for socioeconomic status). Provide clear definitions of the relevant terms used, how they were provided (by the participants/respondents, the researchers, or third parties), and the method(s) used to classify people into the different categories (e.g. self-report, census or administrative data, social media data, etc.) Please provide details about how you controlled for confounding variables in your analyses.* |
| Population characteristics | *Describe the covariate-relevant population characteristics of the human research participants (e.g. age, genotypic information, past and current diagnosis and treatment categories). If you filled out the behavioural & social sciences study design questions and have nothing to add here, write "See above."* |
| Recruitment | *Describe how participants were recruited. Outline any potential self-selection bias or other biases that may be present and how these are likely to impact results.* |
| Ethics oversight | *Identify the organization(s) that approved the study protocol.* |

Note that full information on the approval of the study protocol must also be provided in the manuscript.

# Field-specific reporting

Please select the one below that is the best fit for your research. If you are not sure, read the appropriate sections before making your selection.

☒ Life sciences          ☐ Behavioural & social sciences          ☐ Ecological, evolutionary & environmental sciences

For a reference copy of the document with all sections, see nature.com/documents/nr-reporting-summary-flat.pdf

# Life sciences study design

All studies must disclose on these points even when the disclosure is negative.

| | |
|---|---|
| Sample size | A minimum of three independent biological replicates (n = 3) was used for experiments in this study, including for proteomics, flow cytometry, FISH, chromosome size and imaging analysis. The data were highly consistent between the biological replicates. Chromosome size and fluorescence intensity measurements were based on at least 85 chromosomes for pre-B cells and a minimum of 25 chromosomes for primary fibroblasts (Mof inducible KO and Msl2 KO), collected over three independent experiments. |
| Data exclusions | There was no exclusion/inclusion of samples in the analysis. |
| Replication | Proteomic experiments were conducted with at least three biological replicates. Flow cytometry and chromosome imaging analyses were also performed using a minimum of three biological replicates to ensure reproducibility of key findings. qRT-PCR analyses were carried out in three independent biological replicates, each with technical triplicates to account for pipetting errors. |
| Randomization | Randomization was not relevant to this study as there was no assignment of samples to different experimental groups. |

| Blinding | Blinding was not relevant since there was no assignment of samples to different experimental groups. |

# Reporting for specific materials, systems and methods

We require information from authors about some types of materials, experimental systems and methods used in many studies. Here, indicate whether each material, system or method listed is relevant to your study. If you are not sure if a list item applies to your research, read the appropriate section before selecting a response.

## Materials & experimental systems

| n/a | Involved in the study |
|---|---|
| ☐ | ☒ Antibodies |
| ☐ | ☒ Eukaryotic cell lines |
| ☒ | ☐ Palaeontology and archaeology |
| ☒ | ☐ Animals and other organisms |
| ☒ | ☐ Clinical data |
| ☒ | ☐ Dual use research of concern |
| ☒ | ☐ Plants |

## Methods

| n/a | Involved in the study |
|---|---|
| ☒ | ☐ ChIP-seq |
| ☐ | ☒ Flow cytometry |
| ☒ | ☐ MRI-based neuroimaging |

## Antibodies

| Antibodies used | Primary antibodies used were: H3K27me3-AF647 (clone:C36B11, 12158S, Cell Signalling), macroH2A1 (ab37264, Abcam), H4K16ac-AF488 (clone: E2B8W, 56999, Cell Signalling), H3K14ac-AF488 (clone: EP964Y, ab277918, Abcam), H4ac-AF488 (acetyl K5 + K8 + K12 + K16) (clone: EPR16606, ab223995, Abcam), Trim28 (ab10484, Abcam, lot:GR288493-50), Kat7 (ab70183, Abcam, lot:GR3364428-11), Noc2L (PA5-101730, Invitrogen), H3K14ac (clone: EP964Y, ab52946, Abcam), H4K16ac (ab109463, Abcam), H4ac (06-598, Merck Millipore, lot: 3473490). Dilutions used for the antibodies are provided in the manuscript. Secondary antibodies used were: goat anti-rabbit Alexa Fluor 568, A-11011, Invitrogen, goat anti-rabbit Alexa 488, A11034, Invitrogen. |
|---|---|
| Validation | H3K27me3-AF647 (clone:C36B11, 12158, Cell Signalling), wide species reactivity expected including mouse, tested for use in flow cytometry and ICC/IF on supplier's website.<br>MacroH2A1 (ab37264, Abcam), reacts with human and mouse samples, tested for use in ICC/IF and knockout-validated on supplier's website.<br>H4K16ac-AF488 (clone: E2B8W, 56999, Cell Signalling), wide species reactivity expected including mouse, tested for use in flow cytometry and ICC/IF on supplier's website. Validated in this study for flow cytometry using conditional Mof-KO mouse fibroblasts.<br>H3K14ac-AF488 (clone: EP964Y, ab277918, Abcam), the unconjugated version of this antibody (same clone) reacts with mouse, rat and human, tested on supplier's website for several applications including chIP and ICC/IF and predicted for use in flow cytometry. Validated in this study for flow cytometry using mouse pre-B cells treated with Kat7 histone acetyltransferase inhibitor.<br>H4ac-AF488 (acetyl K5 + K8 + K12 + K16) (clone: EPR16606, ab223995, Abcam), the unconjugated version of this antibody (same clone) reacts with mouse, rat and human samples, validated on supplier's website for ICC/IF.<br>Trim28 (ab10484, Abcam), reacts with mouse and human samples, validated for use in IHC on supplier's website.<br>Kat7 (ab70183, Abcam), reacts with mouse, rat and human samples and validated for use in ICC/IF on supplier's website.<br>Noc2L (PA5-101730, Invitrogen), reacts with mouse, rat and human samples and validated for ICC/IF on supplier's website.<br>H3K14ac (clone: EP964Y, ab52946, Abcam), reacts with mouse, rat and human, tested on supplier's website for several applications including chIP and ICC/IF.<br>H4K16ac (ab109463, Abcam), reacts with mouse, rat and human samples and validated for ICC/IF on supplier's website.<br>H4ac (06-598, Merck Millipore), wide species reactivity predicted and tested for ICC/IF on supplier's website. |

## Eukaryotic cell lines

Policy information about cell lines and Sex and Gender in Research

| Cell line source(s) | Abelson-transformed pre-B cell lines were previously generated in our lab (Lavagnolli et.al, Genes Dev 2015). Mouse Embryonic Fibroblasts (MEFs) used in this study were Mof fl/fl Caag ERT2-cre MEFs and Msl2 KO or WT MEFs, from Asifa Akhtar's lab (Sheikh et al, Oncogene 2016; Sun et al, Nature 2023). |
|---|---|
| Authentication | All cell lines were tested for karyotype and genotyped using appropriate primers. |
| Mycoplasma contamination | All cell lines were tested negative for mycoplasma contamination. |
| Commonly misidentified lines<br>(See ICLAC register) | No commonly misidentified lines were used in this study |

# Plants

| | |
|---|---|
| Seed stocks | *Report on the source of all seed stocks or other plant material used. If applicable, state the seed stock centre and catalogue number. If plant specimens were collected from the field, describe the collection location, date and sampling procedures.* |
| Novel plant genotypes | *Describe the methods by which all novel plant genotypes were produced. This includes those generated by transgenic approaches, gene editing, chemical/radiation-based mutagenesis and hybridization. For transgenic lines, describe the transformation method, the number of independent lines analyzed and the generation upon which experiments were performed. For gene-edited lines, describe the editor used, the endogenous sequence targeted for editing, the targeting guide RNA sequence (if applicable) and how the editor was applied.* |
| Authentication | *Describe any authentication procedures for each seed stock used or novel genotype generated. Describe any experiments used to assess the effect of a mutation and, where applicable, how potential secondary effects (e.g. second site T-DNA insertions, mosiacism, off-target gene editing) were examined.* |

# Flow Cytometry

## Plots

Confirm that:

☒ The axis labels state the marker and fluorochrome used (e.g. CD4-FITC).

☒ The axis scales are clearly visible. Include numbers along axes only for bottom left plot of group (a 'group' is an analysis of identical markers).

☐ All plots are contour plots with outliers or pseudocolor plots.

☒ A numerical value for number of cells or percentage (with statistics) is provided.

## Methodology

| | |
|---|---|
| Sample preparation | Chromosome sorting: Chromosomes were extracted from pre-B cell lines or MEFs and stained with Hoechst 33258 and Chromomycin A3. Individual chromosome populations were purified using BD Influx with BD FACS software (v1.2.0.142). Hoechst 33258 was excited using a 355 nm laser (350mW) and fluorescence was collected using a 400 nm long pass filter in combination with a 500 nm short pass filter. Chromomycin A3 was excited using a 457 nm laser (300mW) and fluorescence was collected using a 500 nm long pass filter in combination with a 600 nm short pass filter. For isolating Xi and Xa, H3K27me3-AF647 was excited using a 637 nm laser (160 mW), and the resulting fluorescence was collected using a 660/30 bandpass filter. Forward scatter was measured using a 488 nm laser (200 mW). Chromosomes were sorted using a 70 μm nozzle tip, with a drop drive frequency of ~96 kHz and sheath pressure at 65 PSI and were collected into FACS tubes containing polyamine buffer.<br>Intracellular staining: For intracellular staining of acetylated histones, cells were fixed and permeabilised using the BD Cytofix/Cytoperm Fixation/Permeabilization Kit according to manufacturer's instructions. Cells were then incubated with conjugated antibodies diluted in permeabilisation buffer for 1 h at RT. After washes, cells were resuspended in FACS buffer (PBS containing 2% FBS) for analysis. |
| Instrument | Chromosomes: BD Influx equipped with spatially separated air cooled lasers. Intracellular staining: BD FACSymphony flow cytometer. |
| Software | BD FACS software (vl.2.0.142, Influx); BD DIVA (v9.1, Symphony) |
| Cell population abundance | Gating strategy used for sorting chromosome X (Xi or Xa), 19, 3 is provided in the manuscript. Sort purity of individual chromosomes was assessed by DNA-FISH with mouse chromosome-specific paints. 80-95% sort purity was achieved. |
| Gating strategy | Chromosomes were first gated on a plot of high Hoechst 33258 vs low Forward scatter signal to gate out debris and clumps. This first gate was then used to create a chromosome karyotype by plotting Hoechst 33258 vs Chromomycin A3 fluorescence. For analysis of cells, gates and voltage were set by first running unstained samples. Cells were first gated based on forward scatter and side scatter and then gated based on side scatter height and area to exclude doublets.<br>Gating strategies used for chromosome sorting/analysis as well as histone acetylation staining in cells are provided in the Supplementary Information File. |

☒ Tick this box to confirm that a figure exemplifying the gating strategy is provided in the Supplementary Information.

