## [Peer Review File · Nature Cell Biology]

Hbo1 and Msl complexes preserve differential compaction and H3K27me3 marking of active and inactive X chromosomes during mitosis

Corresponding Author: Professor Amanda Fisher

Version 0:

Decision Letter:

Revise extended OD

*Please delete the link to your author homepage if you wish to forward this email to co-authors.

Dear Professor Fisher,

"Please accept our sincerest apologies for the length of time your manuscript has been under consideration at our journal. This is because we have been persistently chasing one referee for the report but unfortunately we haven't received it yet. We will forward the report if we receive it in the future. Please note that if/when we do receive the comments of the missing referee, we will pass them on to you and we would expect that the revised manuscript addresses all of the referees' concerns, including those sent subsequently.

Your manuscript, "Hbo1 and Msl complexes preserve differential compaction and H3K27me3 marking of active and inactive X chromosomes during mitosis", has now been seen by 2 referees, who are experts in X chromosome inactivation (referee 1); and chromosome folding (referee 2). As you will see from their comments (attached below) they find this work of potential interest, but have raised substantial concerns, which in our view would need to be addressed with considerable revisions before we can consider publication in Nature Cell Biology.

Nature Cell Biology editors discuss the referee reports in detail within the editorial team, including the chief editor, to identify key referee points that should be addressed with priority, and requests that are overruled as being beyond the scope of the current study. To guide the scope of the revisions, I have listed these points below. We are committed to providing a fair and constructive peer-review process, so please feel free to contact me if you would like to discuss any of the referee comments further.

In particular, it would be essential to:

A- Perform further experiments to test whether the reported effects are specific to Xa or are also observed in autosomes (as per Reviewer#1's points)

B- Extend the comparison of the Xa to the Xi and of both to the autosomes, as per Reviewer#1.

C- Add discussion and interpretation of the Msl/Mof genetic ablation experiment, loss of acetylation and impact on H3K27me3 and X chromosome composition, and the differences in H3K27me3 levels observed between Hbo1 inhibition and Msl/Mof genetic ablation (Reviewer#3)

D- Expand the functional analyses as per Rev#X's paragraph 'As mentioned above, the new insights...' (Reviewer#3)

E- Experimentally test whether TRIM28 is enriched onto the interphase Xi (Reviewer#1)

F- All other referee concerns pertaining to strengthening existing data, providing controls, methodological details, clarifications and textual changes, should also be addressed.

G- Finally please pay close attention to our guidelines on statistical and methodological reporting (listed below) as failure to do so may delay the reconsideration of the revised manuscript. In particular please provide:

We would be happy to consider a revised manuscript that would satisfactorily address these points, unless a similar paper is published elsewhere, or is accepted for publication in Nature Cell Biology in the meantime.

- ensure that it conforms to our format instructions and publication policies (see below and www.nature.com/nature/authors/).

- provide a point-by-point rebuttal to the full referee reports verbatim, as provided at the end of this letter.

- provide the completed Editorial Policy Checklist (found here <https://www.nature.com/authors/policies/Policy.pdf>), and Reporting Summary (found here <https://www.nature.com/authors/policies/ReportingSummary.pdf>). This is essential for reconsideration of the manuscript and these documents will be available to editors and referees in the event of peer review. For more information see <http://www.nature.com/authors/policies/availability.html> or contact me.

Nature Cell Biology is committed to improving transparency in authorship. As part of our efforts in this direction, we are now requesting that all authors identified as 'corresponding author' on published papers create and link their Open Researcher and Contributor Identifier (ORCID) with their account on the Manuscript Tracking System (MTS), prior to acceptance. ORCID helps the scientific community achieve unambiguous attribution of all scholarly contributions. You can create and link your ORCID from the home page of the MTS by clicking on 'Modify my Springer Nature account'. For more information please visit <http://www.springernature.com/orcid>.

Link Redacted

We would like to receive a revised submission within six months. We would be happy to consider a revision even after this timeframe, however if the resubmission deadline is missed and the paper is eventually published, the submission date will be the date when the revised manuscript was received.

We hope that you will find our referees' comments, and editorial guidance helpful. Please do not hesitate to contact me if there is anything you would like to discuss.

Best wishes,

Sabrya Carim

Sabrya Carim, PhD
(she/her/hers)
Associate Editor, Nature Cell Biology
Nature Portfolio

Springer Nature
The Campus, 4 Crinan Street, London N1 9XW, UK
sabrya.carim@springernature.com
<https://orcid.org/0000-0001-9485-1938>

Reviewers' Comments:

Reviewer #1:

Remarks to the Author:

This study further developed a previously established chromosome sorting strategy to enable the isolation of the active and inactive X chromosome from pre-B cells utilizing the difference in H3K27me3 to distinguish the Xi and Xa. The faithful isolation of the Xi and Xa was nicely confirmed by DNA methylation analysis. Imaging analysis shows that the mitotic Xi is more compact (smaller) than the Xa or autosomes. Thus, the compaction of the X is maintained throughout mitosis. The authors also confirm that various histone acetylation marks are depleted on the mitotic Xi compared to the mitotic Xa, and H3K27me3 enriched on the Xi, which is well established in the field (although maybe not for all acetylation marks shown here and not by mass spectrometry). It remains unclear if Xist is still associated with the sorted chromosomes.

Exploiting this technology, the authors applied proteomics to the mitotic Xa and Xi in female cells, the mitotic Xa in male cells and the autosomes 3 and 19 in bot. They make various comparisons between these chromosomes in protein composition which represent a resource for the field. The finding the authors focus on is that the Xa is enriched for the Hbo1 and MSL histone acetyltransferases. The authors suggest that this enrichment is present specifically on the female Xa and not on the male Xa and not on autosomes. However, there is no explanation for this interesting finding. The authors then suggest that inhibition/deletion Hbo1 or Msl2 induce changes in H3K27me3 on the X chromosomes and changes in X chromosome compaction. However, for the Msl2 KOs these data are not split by Xa/Xi it therefore remains unclear what exactly is happening. Moreover, autosomes are also changing, albeit different autosomes (3 and 19) in different directions, raising the question of how these findings matter for the Xa and Xi versus the autosomes, and indicating that

there are global changes in the nucleus.

Overall, this is an extensive and well-written study that provides a new and exciting method to purify the mitotic Xa and Xi. The data provide further insights into the proteomic differences between the Xi and Xa, but the functional studies on Msl2 do not distinguish the Xi from the Xa, and the inhibition of HBO1 and deletion of Msl2 also affects autosomal size, although different autosomes are affected differently. Thus, whether this regulation is specific to the Xa and Xi interaction is unclear. I have specific points that should be addressed, if possible, with a specific focus of extending the comparison of the Xa to the Xi and of both to various autosomes (chr3 and chr19). It is really not clear to me whether the Xa is special. Maybe the focus on Xi/Xa crosstalk should be reduced.

Specific Points:

Generally, I wonder whether the isolated inactive X chromosome is associated with Xist RNA. It will be good to perform imaging for Xist or an RT-PCR. Is Xist enriched one would expect the enrichment of XIST-interacting RNA-binding proteins?

Chromosome 19 is relatively unique in its gene content with large Znf gene regions. It would be nice to see the data for another chromosome as control, or all autosomes, or the most X-chromosome-like chromosome in gene density for most plots.

The conclusion of Figure 1 "These data indicated that although the representation of chromosome-associated RNA-binding proteins at metaphase was different in female and male pre-B cells, there was no evidence implicating Xist-interacting factors in the preservation of XCI through mitosis." may be fine but the sensitivity of the approach comparing female and male cells might be too low. Plus, many of the RNA-binding proteins are also present on chromosome 19, thus the results observed here are not X-chromosome-specific.

For the mass spec analysis in Figure 2G it is also unclear how many replicates were analyzed per condition. Is the PCA plot in Figure S2c indicating the number of replicates?

It would be nice to include an active autosome in the analysis in Figure S2c (for the Xi and Xa-specific MS analysis).

There are extensive differences between the Xa in female cells versus the Xa in male cells. What does this mean? (Supp Fig 2D).

The authors should show whether TRIM28 is enriched on the interphase Xi. Otherwise it would not be clear why it is pointed out that it does not enrich on the mitotic Xi.

Are the histone acetylation proteins also detected on the autosomes or specific to the Xa? This remains somewhat unclear to me. Figure 3a-c should compare the Xi to an autosome and the autosome to the Xa. I appreciate Figure 3e showing differences of some proteins of the Xa to autosomes and the male Xa, however, it would be better to show the plots as in 3a-c for the Xa to Xi to autosome comparisons, since the authors argue that the Xa is different from autosomes. Moreover, again, chromosome 19 might not be the best since it has various heterochromatic Znf regions. This is important since chr3 was also purified and the data could be shown. Similarly, MEAF6 is not significant between the Xi and Xa but compared to chr 19 in Figure 3e. For the histone mark comparison in Fig 3f, it would be important to show controls – such as H3K27me3 and H3K9me2/3 as they should be enriched on the Xi. The comparison of the histone marks should ideally also be shown to an autosome (different from chromosome 19) to understand if the Xa is different. Regardless, it is well known that the Xi is depleted of histone acetylation relative to the Xa in mitosis, but this may not have been shown for H3K14ac yet.

For the Xi accumulation of the proteins Noc2l and Mybbp1 it would also be nice to see the entire plot of proteins against the male Xa and autosomes. What complexes are these proteins in? It would be interesting to see the staining for these proteins on complete mitotic and interphase cells in comparison to Xist.

The authors perform two functional experiments. First, they inhibit Kat7 chemically to deplete H3K14ac. Surprisingly, the size of the Xa is reduced, yet autosomes behave differently with chr19 reduced in size and chr3 unchanged. The Xi also does not change. The authors suggest that the Xa reaches the size of the Xi. However, chr3 changes as well. Thus, this affect is not specific to the Xa and likely affects many chromosomes that are actively expressed.

Second, the authors delete Msl2 using direct and conditional KO MEFs. In Figure 5b it is not clear which H3K27me3 data in MSL2 KO MEFs represent the Xa and Xi. Is the level of H3K27me3 on the Xi reduced and on the Xa increased? How does this look by imaging in mitotic cells directly? On the autosomes, significant changes are also observed, going into distinct direction on Chr3 and Chr19. This result is understated in the manuscript, and it seems many chromosomes change, not just the Xa.

The chromosome size measurement in Fig 5c compares the Xa and Xi in wt cells with the total X measurements in MSL2 KO Mefs. These data should be split for the Xa and Xi, are both reduced in size?

The H3K27me3 data in conditional MEFs should be split by Xa and Xi and quantified for these chromosomes as well as for chr 3 and chr 19 in boxplots for an easier comparison. It is interesting that the size of the X chromosome is more dramatically affected in conditional Msl2 MEFs than the size of chromosomes 3 and 19. Can this be split into Xa and Xi – are both reduced in size? The size difference of the total X compared to the Xi and Xa in wt MEFs is more pronounced in conditional MEFs than KO MEFs, which suggests both X chromosomes are affected?

Are these findings regarding H3K27me3 and chromosome size for mitotic chromosomes of MSL2 KO cells reproduced in the absence of sorting by simple imaging?

Minor Points

In Figure 1a, the BARR body may be better demonstrated by showing the dapi-intense body (ie show the dapi channel as well which is currently not shown but mentioned in the text).

How many replicates were done for the MS experiment in Figure 1.

Chromosome 19 is relatively unique in its gene content with large Znf gene regions. It would be nice to see the data for another chromosome as control, or all autosomes, or the most X-chromosome-like chromosome in gene density.

Significance tests should be added to Figure 2f.

The dot-plot analysis for macroH2A in Figure 2d is missing.

Reviewer #2:

Remarks to the Author:

Summary:

Here, Djeghoul et al. expand on their chromosome flow-sorting method to isolate and characterize active (Xa) versus inactive X (Xi) chromosomes from mitotic pre-B cells and primary fibroblasts. Expectedly, mitotic Xi chromosomes were more compact and more enriched for H3K27me3 compared to mitotic Xa chromosomes. Proteomics analysis of isolated Xi versus Xa chromosomes revealed an unexpected and interesting role of histone acetylation in Xi versus Xa chromosome structure, observations that are supported by pharmacological (Hbo1 inhibition) and genetic (Msl/Mof ablation) experiments. As the authors point out, and this reviewer agrees, the ability to isolate individual native mitotic chromosomes from a variety of biological samples is a very novel and very exciting technology. The approach is valid, the data of high quality and analyzed appropriately, and the manuscript is mostly well written with clear figures. The new insights regarding acetylation and X chromosome biochemistry and structure are also important additions for the X inactivation field. The comments below are intended to further strengthen and improve the manuscript prior to publication.

Major comments:

The paper would likely benefit from a bit more discussion and interpretation from experiments that caused loss of acetylation due to Msl/Mof genetic ablation and its impact on chr X composition in relation to H3K27me3. The intermediate levels of H3K27me3 are interesting, and, as the authors point out, the mechanism is not fully worked out. Are these intermediate values due to both loss H3K27me3 on chr Xi and gain of H3K27me3 on chr Xa?

Related to the above comment, is it possible that H3K27ac levels could change 4-6 passages after Msl/Mof genetic ablation? If so, could a change in the balance of H3K27ac and H3K27me3 explain some of the results – i.e. if H3K27 is not acetylated it can then be methylated resulting in increased H3K27me3 on chr Xa. The authors see differences in H3K27ac levels on Xi versus Xa chromosomes (Fig 3f) so investigating H3K27ac levels by flow cytometry when H3K27me3 levels change upon Msl/Mof depletion may help to elucidate this mechanism.

Also related to the first comment, do the authors have any insight into why Hbo1 inhibition by WM-3835 does not cause changes in H3K27me3 levels whereas Msl/Mof genetic ablation does?

The CpG methylation analysis convincingly demonstrates that the structure of the isolated X chromosomes is maintained after flow sorting. Nonetheless, quantifying the enrichment of the isolated X chromosomes would be beneficial. This could be performed by qPCR (or shallow whole-genome sequencing, though this is more involved and costly) for a few loci from the isolated X chromosomes from total extracted chromosomes versus flow-sorted chromosomes compared to the same analysis for an autosome. This would provide the reader with how much they have enriched the Xi and Xa chromosomes, thereby providing an indication of the purity of the isolated chromosomes.

The authors performed the control experiment for loss of H3K14ac staining after WM-3835 inhibitor treatment on whole cells (Fig. 4d, top panel). Given the focus of this manuscript, it would be more appropriate to assess this for the flow sorted chr X to confirm a direct effect on chr X.

As mentioned above, the new insights into acetylation and X chromosome biochemistry and structure are interesting and important for X inactivation. The paper would benefit from a bit of functional data in this regard. Have the authors performed RNA-seq after Hbo1 inhibition and/or Msl/Mof genetic ablation? If so, does the expression of genes on the chr X change in a way that is consistent with changes in X inactivation?

Minor comments:

Line 154 states “2562 and 2375 protein hits differentially detected”. Do the authors mean that 2562 and 2375 proteins in total were detected? Based on the numbers in red and blue in Fig. 1E it looks as though 176 and 281 differential proteins are detected.

Comparing line 173 with Fig. S1E, is the number of proteins enriched only on male chr X 43 or 33?

Comparing line 345 with Fig. 5F it is unclear how chr X purity is quantified.

Reviewer #3:

None

Methods should be written concisely, but should contain all elements necessary to allow interpretation and replication of the results. As a guideline, Methods sections typically do not exceed 3,000 words. The Methods should be divided into subsections listing reagents and techniques. When citing previous methods, accurate references should be provided and any alterations should be noted. Information must be provided about: antibody dilutions, company names, catalogue numbers and clone numbers for monoclonal antibodies; sequences of RNAi and cDNA probes/primers or company names and catalogue numbers if reagents are commercial; cell line names, sources and information on cell line identity and authentication. Animal studies and experiments involving human subjects must be reported in detail, identifying the committees approving the protocols. For studies involving human subjects/samples, a statement must be included confirming that informed consent was obtained. Statistical analyses and information on the reproducibility of experimental results should be provided in a section titled "Statistics and Reproducibility".

All Nature Cell Biology manuscripts submitted on or after March 21 2016 must include a Data availability statement at the end of the Methods section. For Springer Nature policies on data availability see <http://www.nature.com/authors/policies/availability.html>; for more information on this particular policy see <http://www.nature.com/authors/policies/data/data-availability-statements-data-citations.pdf>. The Data availability statement should include:

- Accession codes for primary datasets (generated during the study under consideration and designated as "primary accessions") and secondary datasets (published datasets reanalysed during the study under consideration, designated as "referenced accessions"). For primary accessions data should be made public to coincide with publication of the manuscript. A list of data types for which submission to community-endorsed public repositories is mandated (including sequence, structure, microarray, deep sequencing data) can be found here <http://www.nature.com/authors/policies/availability.html#data>.
- Unique identifiers (accession codes, DOIs or other unique persistent identifier) and hyperlinks for datasets deposited in an approved repository, but for which data deposition is not mandated (see here for details <http://www.nature.com/sdata/data-policies/repositories>).

- At a minimum, please include a statement confirming that all relevant data are available from the authors, and/or are included with the manuscript (e.g. as source data or supplementary information), listing which data are included (e.g. by figure panels and data types) and mentioning any restrictions on availability.
- If a dataset has a Digital Object Identifier (DOI) as its unique identifier, we strongly encourage including this in the Reference list and citing the dataset in the Methods.

We recommend that you upload the step-by-step protocols used in this manuscript to protocols.io. More details can be found at <https://www.protocols.io/help/publish-articles>.

All imaging data should be accompanied by scale bars, which should be defined in the legend. Cropped images of gels/blots are acceptable, but need to be accompanied by size markers, and to retain visible background signal within the linear range (i.e. should not be saturated). The boundaries of panels with low background have to be demarked with black lines. Splicing of panels should only be considered if unavoidable, and must be clearly marked on the figure, and noted in the legend with a statement on whether the samples were obtained and processed simultaneously. Quantitative comparisons between samples on different gels/blots are discouraged; if this is unavoidable, it should only be performed for samples derived from the same experiment with gels/blots were processed in parallel, which needs to be stated in the legend.

- For line art, graphs, charts and schematics we prefer Adobe Illustrator (.AI), Encapsulated PostScript (.EPS) or Portable Document Format (.PDF). Files should be saved or exported as such directly from the application in which they were made, to allow us to restyle them according to our journal house style.
- We accept PowerPoint (.PPT) files if they are fully editable. However, please refrain from adding PowerPoint graphical effects to objects, as this results in them outputting poor quality raster art. Text used for PowerPoint figures should be Helvetica (preferred) or Arial.
- We do not recommend using Adobe Photoshop for designing figures, but we can accept Photoshop generated (.PSD or .TIFF) files only if each element included in the figure (text, labels, pictures, graphs, arrows and scale bars) are on separate layers. All text should be editable in 'type layers' and line-art such as graphs and other simple schematics should be preserved and embedded within 'vector smart objects' - not flattened raster/bitmap graphics.
- Some programs can generate Postscript by 'printing to file' (found in the Print dialogue). If using an application not listed above, save the file in PostScript format or email our Art Editor, Allen Beattie for advice (a.beattie@nature.com).

The total number of Supplementary Figures (not including the "unprocessed scans" Supplementary Figure) should not exceed the number of main display items (figures and/or tables (see our Guide to Authors and March 2012 editorial <http://www.nature.com/ncb/authors/submit/index.html#suppinfo>; <http://www.nature.com/ncb/journal/v14/n3/index.html#ed>). No restrictions apply to Supplementary Tables or Videos, but we advise authors to be selective in including supplemental data.

GUIDELINES FOR EXPERIMENTAL AND STATISTICAL REPORTING

REPORTING REQUIREMENTS – To improve the quality of methods and statistics reporting in our papers we have recently revised the reporting checklist we introduced in 2013. We are now asking all life sciences authors to complete two items: an Editorial Policy Checklist (found here <https://www.nature.com/authors/policies/Policy.pdf>) that verifies compliance with all required editorial policies and a reporting summary (found here <https://www.nature.com/authors/policies/ReportingSummary.pdf>) that collects information on experimental design and reagents. These documents are available to referees to aid the evaluation of the manuscript. Please note that these forms are dynamic 'smart pdfs' and must therefore be downloaded and completed in Adobe Reader. We will then flatten them for ease of use by the reviewers. If you would like to reference the guidance text as you complete the template, please access these flattened versions at <http://www.nature.com/authors/policies/availability.html>.

Version 1:

Decision Letter:

Our ref: NCB-A54772A

1st May 2025

Dear Dr. Fisher,

Thank you for submitting your revised manuscript "Hbo1 and Msl complexes preserve differential compaction and H3K27me3 marking of active and inactive X chromosomes during mitosis" (NCB-A54772A) and for your patience with the review process. It has now been seen by the original referees and their comments are below. The reviewers find that the paper has improved in revision, and therefore we'll be happy in principle to publish it in Nature Cell Biology, pending minor revisions to satisfy the referees' final requests and to comply with our editorial and formatting guidelines.

Please ensure that all figures fit into a single page (not multiple pages) and adhere to a maximum page size of roughly 180mm wide x 200mm high and use a font size of no smaller than 6pt Arial or Helvetica throughout the figures, to ensure legibility of the figures once resized for publication. We encourage you to make these changes in font size at this step, to avoid any potential delays in the production process.

We are now performing detailed checks on your paper and will send you a checklist detailing our editorial and formatting requirements in about a week. **Please do not upload the final materials and make any revisions until you receive this additional information from us.**

Thank you again for your interest in Nature Cell Biology Please do not hesitate to contact me if you have any questions.

Best wishes,

Sabrya Carim, PhD
(she/her/hers)
Senior Editor, Nature Cell Biology
Nature Portfolio

Springer Nature
The Campus, 4 Crinan Street, London N1 9XW, UK
sabrya.carim@springernature.com
<https://orcid.org/0000-0001-9485-1938>

Reviewer #1 (Remarks to the Author):

In this revision, the authors performed additional experiments and clarified the text. I appreciate the effort and the quality of the work. Overall, using chromosome sorting, the authors find that the Xi carries high H3K27me3 levels and is compacted compared to the Xa. This is well established. However, I do like the chromosome sorting combined with proteomics and the availability of this data resource. However, this finding is expected from decades of research on the inactive X chromosome. The authors argue in the rebuttal letter:

“Previous proteomic analyses of mitotic chromosomes purified en masse from mouse ESCs (Djeghloul et al., 2020 PMID: 32807789, Djeghloul et al., 2023 PMID: 36941433) revealed that components of repressive chromatin complexes (such as PRC1, PRC2 and DNA methyltransferases) co-enrich with metaphase chromosomes, whereas generally, HAT complexes did not. In view of this, it was unexpected that in mouse pre-B cells, HAT complex components showed enrichment on the Xa as compared to the Xi (top panel, Fig. 3a), or as compared to an autosome (mouse chromosome 19, Fig. 3d). These data suggest that active X chromosomes isolated from males or females, show a propensity to retain these HAT complexes through mitosis.”

If this is the central new finding of this study, the manuscript should be reframed to make that clearer, as it didn't come through to me until Figure 3 and the discussion. The Xi-related work, as currently discussed, seems somewhat limited in terms of novel insight and the authors argue themselves that the central finding relates to the acetylation.

Reviewer #2 (Remarks to the Author):

Although the mechanism by which loss of Msl2/Mof leads to changes in H3K27me3 levels is not yet understood, the authors clearly state this in the manuscript. All of my other concerns have been addressed by the authors.

Reviewer #3 (Remarks to the Author):

The authors have very carefully addressed most of my comments, including important new data, corrected concepts and modified text to more adequately described the results and some important limitations. The differential apparent condensation(size) of distinct chromosomes not following the expected trend upon drug treatment or genetic manipulation remains an intriguing result. As stated before, the experimental approach is very impressive and provided striking new results leading to important new concepts about the maintenance of the active X chromosome through mitosis. This opens the door to exciting follow-up lines of investigation. It certainly deserves publication in NCB.

Version 2:

Decision Letter:

Dear Dr Fisher,

I am pleased to inform you that your manuscript, "Hbo1 and Msl complexes preserve differential compaction and H3K27me3 marking of active and inactive X chromosomes during mitosis", has now been accepted for publication in Nature Cell Biology. Many congratulations!

Please note that *Nature Cell Biology* is a Transformative Journal (TJ). Authors may publish their research with us through the traditional subscription access route or make their paper immediately open access through payment of an article-processing charge (APC). Authors will not be required to make a final decision about access to their article until it has been accepted. [Find out more about Transformative Journals](https://www.springernature.com/gp/open-research/transformative-journals)

Authors may need to take specific actions to achieve compliance with funder and institutional open access mandates. If your research is supported by a funder that requires immediate open access (e.g. according to [Plan S principles](https://www.springernature.com/gp/open-science/plan-s-compliance) or the [NIH public access policy](https://www.springernature.com/gp/open-science/us-federal-agency-compliance)) then you should select the gold OA route, and we will direct you to the compliant route where possible. Because authors warrant under our subscription licensing terms that they haven't committed to licensing any version of their article under a licence inconsistent with the terms of our agreement – including the applicable embargo period – publication under the subscription model isn't suitable for authors whose funders require no embargo.

If you have not already done so, we strongly recommend that you upload the step-by-step protocols used in this manuscript to protocols.io (<https://protocols.io>), an open online resource that allows researchers to share their detailed experimental know-how. All uploaded protocols are made freely available and are assigned DOIs for ease of citation. Protocols and Nature Portfolio journal papers in which they are used can be linked to one another, and this link is clearly and prominently visible in the online versions of both. Authors who performed the specific experiments can act as primary authors for the Protocol as they will be best placed to share the methodology details, but the Corresponding Author of the present research paper should be included as one of the authors. By uploading your Protocols onto protocols.io, you are enabling researchers to more readily reproduce or adapt the methodology you use, as well as increasing the visibility of your protocols and papers. You can also establish a dedicated workspace to collect your lab Protocols. Further information can be found at <https://www.protocols.io/help/publish-articles>.

Nature Cell Biology encourages authors presenting evidence for cell, biological, molecular, and genetic interactions to consider communicating these findings using Biofactoid (<https://biofactoid.org/>). This tool helps users share a searchable representation of interactions (e.g. binding, gene expression, post-translational modification) between genes, gene products, or chemicals. Information

added to Biofactoid, with author attribution, is shared on social media and public databases, such as Pathway Commons, where it can be discovered and analyzed in the context of a large and growing corpus of knowledge.

With kind regards,

Sabrya.

Sabrya Carim, PhD
(she/her/hers)
Senior Editor, Nature Cell Biology
Nature Portfolio

Springer Nature
The Campus, 4 Crinan Street, London N1 9XW, UK
sabrya.carim@springernature.com
<https://orcid.org/0000-0001-9485-1938>

** Visit the Springer Nature Editorial and Publishing website at http://editorial-jobs.springernature.com?utm_source=ejp_NCB_email&utm_medium=ejp_NCB_email&utm_campaign=ejp_NCB for more information about our career opportunities. If you have any questions please click [here](mailto:editorial.publishing.jobs@springernature.com).
